# Air-borne in-situ measurements of aerosol size distributions and BC across the IGP during SWAAMI -RAWEX

Mukunda Madhab Gogoi[1], Venugopalan Nair Jayachandran[1], Aditya Vaishya[2], Surendran Nair Suresh Babu[1], Sreedharan Krishnakumari Satheesh[3,4]and Krishnaswamy Krishna Moorthy[3]

[1]Space Physics Laboratory, Vikram Sarabhai Space Centre, Thiruvananthapuram – 695022, India
[2]School of Arts and Sciences & Global Centre for Environment and Energy, Ahmedabad University, Ahmedabad – 380009, India
[3]Centre for Atmospheric and Oceanic Sciences, Indian Institute of Science, Bengaluru – 560012, India
[4]Divecha Centre for Climate Change, Indian Institute of Science, Bengaluru – 560012, India

## Abstract

During the combined South-West Asian Aerosol Monsoon Interaction – Regional Aerosol Warming Experiment (SWAAMI -RAWEX), collocated air-borne measurements of aerosol number-size distributions in the size (diameter) regime 0.5 to 20 µm and black carbon (BC) mass concentrations were made across the Indo-Gangetic Plains (IGP), for the first time, from three distinct locations, just prior to the advent of Indian Summer Monsoon over the IGP. These measurements provided an east-west transect of region-specific properties of aerosols as the environment transformed from mostly-arid conditions of western IGP (represented by Jodhpur, JDR) having dominance of natural aerosols to the Central IGP (represented by Varanasi, VNS) having very high anthropogenic emissions, to the eastern IGP (represented by the coastal station Bhubaneswar, BBR) characterized by a mixture of the IGP outflow and marine aerosols. Despite these, the aerosol size distribution revealed an increase in coarse mode concentration and coarse mode mass-fraction (fractional contribution to the total aerosol mass) with the increase in altitude across the entire IGP, especially above the well-mixed region. Consequently, both the mode radii and geometric mean radii of the size distributions showed an increase with altitude. However, near the surface and within the atmospheric boundary layer (ABL), the features were specific to the different sub-regions; with the highest coarse mode mass fraction ($F_{MC}$ ~72%) in the western IGP and highest accumulation fraction in the Central IGP with the eastern IGP coming in-between. The elevated coarse mode fraction is attributed to mineral dust load arising from local production as well as due to advection from the west. This was further corroborated by data from Cloud Aerosol Transportation System (CATS) onboard International Space Station (ISS), which also revealed that the vertical extent of dust aerosols reached as high as 5 km during this period. Mass concentrations of BC were moderate (~1 µg m$^{-3}$) with very little altitude variation up to 3.5 km, except over VNS where very high concentrations were seen near the surface and within the ABL. BC induced atmospheric heating rate was highest near the surface at VNS (~ 0.81 K day$^{-1}$), while showing an increasing pattern with altitude at BBR (~ 0.35 K day$^{-1}$ at the ceiling altitude).

Keywords: Aerosol size distribution profile, BC mass fraction, aerosol type, IGP, monsoon.

**Corresponding Author**:
Dr. Mukunda M. Gogoi
Space Physics Laboratory, Vikram Sarabhai Space Centre
Indian Space Research Organization, Thiruvananthapuram – 695022, India
Email: dr_mukunda@vssc.gov.in
Phone: +91-471-256 3365; Fax: +91-471-270 6535

## 1. Introduction

The Indo-Gangetic Plains (IGP) remains one of the global hotspots of aerosols. The prevailing high aerosol loading and the relative abundance of its constituents (being a mixture of natural and anthropogenic species) is known to show significant seasonality (Gautam et al., 2011; Praveen et al., 2012; Moorthy et al., 2016; Vaishya et al., 2018; Rana et al., 2019; Brooks et al., 2019). This arises due to combined effects of the dense population and the associated anthropogenic and industrial activities, as well as the loose alluvial soil of this regions having vast semi-arid and arid characteristics to the west. A dense network of thermal power-plants, several of them being coal fired, is among the prominent source of anthropogenic emissions over the region. This is abetted by the synoptic meteorology with its strong seasonality (Gautam et al., 2010; Nath et al., 2018; Singh et al., 2018) and the orography that slopes down from the west to east bound on the north and south respectively by the Himalayas and the Aravalli ranges and Bihar Plateau forming a confined channel (Moorthy et al. 2007; Gogoi et al., 2017). For accurate quantification of the radiative implications of this complex aerosol system, several concerted studies have been made using ground based (Giles et al., 2012; Bansal et al., 2019) and space-borne measurements (Srivastava, 2016; Mhawish et al., 2017; Kumar et al., 2018) as well as numerical modeling (Govardhan et al., 2019). However, most of these studies have uncertainties arising out of the ill-represented altitude variation of aerosol properties due to sparse measurements. Height resolved in-situ measurements of aerosol properties are indispensable not only in this regard, but also for understanding aerosol-cloud interactions.

In recent years, a few campaign-mode airborne measurements have been made over this region to estimate the altitude-resolved properties of aerosols that are important in aerosol-radiation interactions (Padmakumari et al., 2013; Babu et al., 2016; Nair et al., 2016; Vaishya et al., 2018; Gogoi et al., 2019). These include the measurements of aerosol scattering and absorption coefficients conducted as part of the Regional Aerosol Warming Experiment (RAWEX; Babu et al., 2016) to delineate the spatio-temporal variability in the altitude distribution of aerosol single scattering albedo (SSA) across the IGP during winter and pre-monsoon seasons and aerosol and cloud parameter measurements conducted as part of the Cloud Aerosol Interaction and Precipitation Enhancement Experiment (CAIPEEX; Kulkarni et al., 2012). Some studies have also reported significant contribution of dust and BC to the elevated aerosol load (Praveen et al., 2012; Kedia et al., 2014; Pandey et al., 2016; Li et al., 2016) and their potential role to act as ice nuclei (Padmakumari et al., 2013). However, despite its importance in radiative interactions and CCN activation, the altitude-resolved measurements of aerosol size distribution are extremely sparse, or non-existent, especially just prior to the onset of the Indian Summer Monsoon, when the sources of aerosols, their mixing

and transport pathways are all complex. The information on aerosol size distribution is important for accurately describing the phase function, which describes the angular variation of the scattered intensity. The knowledge of its vertical variation would thus improve the accuracy of ARF estimation and hence heating rates. Such information is virtually non-existing over this region. Further, the knowledge of the variation of size distribution with altitude would be useful in better understanding the aerosol-cloud interactions and CCN characteristics, during the evolving and active phase of the Indian monsoon. This was among the important information aimed to be obtained under SWAAMI - RAWEX (https://gtr.ukri.org/projects?ref=NE%2FL013886%2F1 and http://www.spl.gov.in/SPL/index.php/arfs-research/field-campaigns/asfasf) - a joint Indo-UK field experiment involving airborne measurements using Indian and UK aircrafts during different phases of the Indian monsoon, right from just prior to the onset of monsoon (i.e. in the beginning of June).

During this campaign, vertical profiles of various aerosol parameters have been measured using an instrumented aircraft from three base stations –representing western, central and eastern end of the IGP– during 01 to 20 June 2016, just prior to onset of the Indian summer monsoon. Some important results on the optical and CCN characteristics are already reported (Vaishya et al., 2018; Jayachandran et al 2020). In the present study, we have examined the vertical profiles of aerosol number-size distributions in the size (diameter) regime 0.5 to 20 µm and black carbon (BC) mass concentrations. The results are presented and discussed in the light of other supplementary information.

**2. Experimental Details and database**

2.1 Study region and flight details

The base stations (Figure-1), from where the aircraft operations were carried out, represented distinct regions of the IGP; 'Jodhpur (JDR; 26.25°N, 73.04°E)' in the western IGP is an arid/ semi-arid region with low urban activities, lying downwind the 'Great Indian Desert' to its west (JDR has population density of 161 per sq. km).'Varanasi (VNS; 25.44°N, 82.85°E)' in the central IGP is located downwind of Jodhpur, characterized by extensive anthropogenic activities (automobiles, small and large-scale industries and thermal power plants and wide spread agricultural activities) by its dense population (density 2,399 $km^{-2}$). 'Bhubaneswar (BBR; 20.25°N, 85.81°E)' is an urban location in the eastern IGP (population density of 2131 $km^{-2}$), and experiences the influence of marine aerosol component from the Bay-of-Bengal (~ 50 km away from the base station) in addition to the influence of IGP outflow and local aerosol sources from nearby thermal (coal based) power plants, mining and fertilizer based industries etc. (Panda et al., 2016). The northwestern part of India

has an undulating topography, due to which monsoon currents loose moisture while crossing the
western mountain ranges (Aravalli) and results in dry arid regions (Moorthy et al., 2007). Strong
dust-rising winds are a common feature of the IGP in general and its western parts in particular
during April to July (Banerjee et al. 2019). In the central IGP, VNS and its environs hold largely
even topography, where the Ganga is the principal river. In the eastern IGP, BBR is topographically
decorated with western uplands and eastern lowlands, with hillocks in the western and northern parts.
These base stations, thus provided a west-east cross section of the highly aerosol laden IGP; where
the aerosol characteristics are known to change longitudinally. The spatial map of AOD at 550 nm
(Figure-1) clearly shows the existence of higher aerosol loading (AOD > 0.5) over the observational
site during the study period.

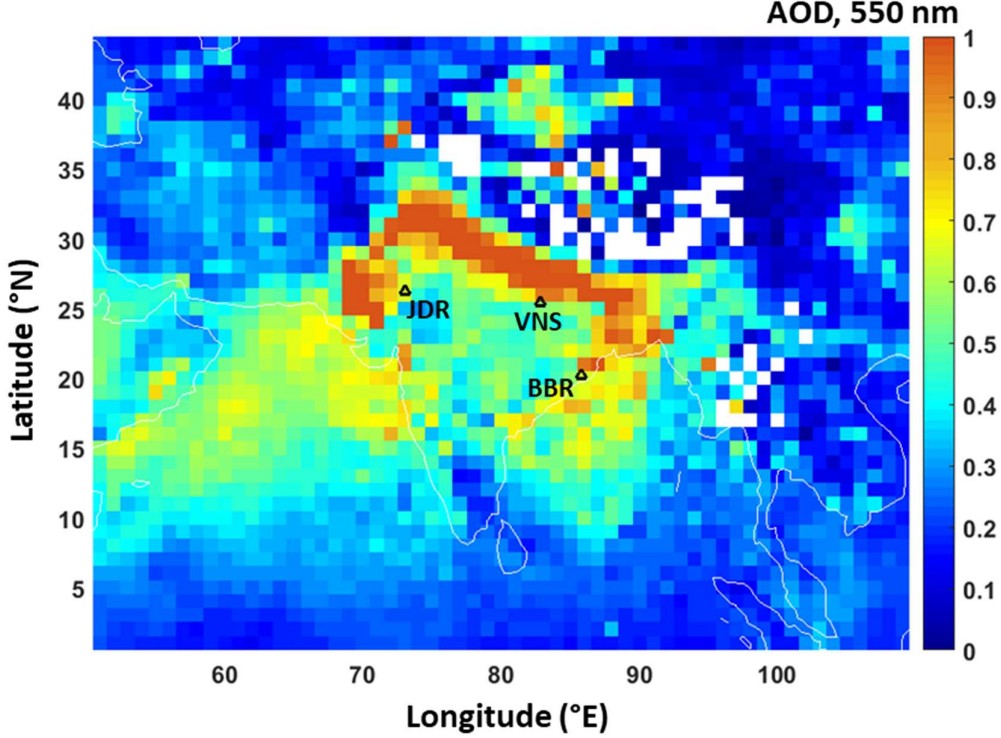


**Figure-1**: Three distinct base stations: (i) 'Jodhpur (JDR; 26.25°N, 73.04°E)' in the western IGP,
(ii) 'Varanasi (VNS; 25.44°N, 82.85°E)' in the central IGP and (iii) 'Bhubaneswar (BBR; 20.25°N,
85.81°E)' in the eastern coastal IGP, from where the aircraft measurements were conducted. The
spatial map of AOD at 550 nm obtained from MODIS sensor (MOD08_D3_6.1, Dark-Target and
Deep-Blue combined mean) on-board Terra satellite during the study period (01-20 June 2016) is
shown in the background.
Figure-2a shows the actual dates of onset of the monsoon at different parts of India in 2016. As can
be seen from the figure, despite a delayed onset at the southern tip of India, monsoon advanced fast
in to the central/northern parts of India. Yet, all the flight sorties from the respective base stations
were completed ahead of the advent of monsoon to that station. At the eastern IGP, the aircraft sorties
were made from 'BBR' before the onset of monsoon over India; at 'VNS', the flights were conducted
while monsoon advanced only to the central peninsula. The final set of sorties were conducted at
'JDR' when the monsoon covered most of the central and eastern part of India, but yet to progress
towards northwestern parts.

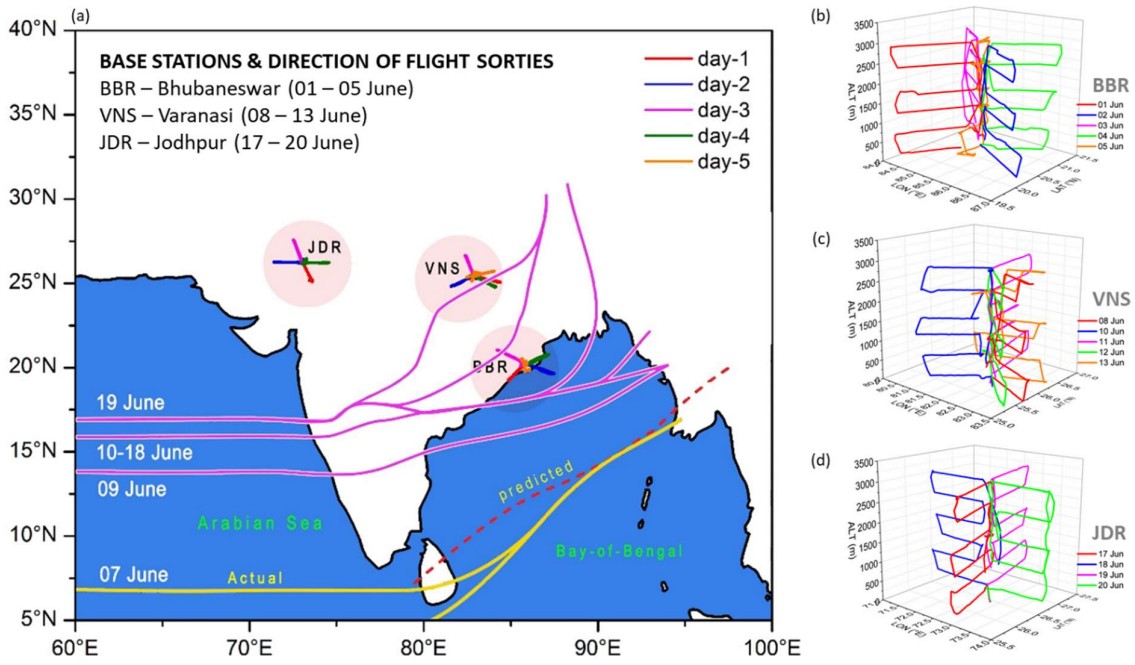


**Figure-2**: (a) The onset (actual) of SW-Monsoon at different parts of India, shown by the yellow
and pink (solid) lines. Horizontal and vertical flight paths during each of the sorties at (b)
Bhubaneswar (BBR), (c) Varanasi (VNS) and (d) Jodhpur (JDR).
From each of the base stations, 4 to 5 sorties were carried out on successive days in different
horizontal directions about the station, as shown by the ground projections (horizontal lines in Fig
2a), with a view to obtain an average sub-regional representation in the shortest time possible. During
each of the sorties, measurements were made at six discrete levels following a staircase configuration
as shown in Figure 2 b-d (for JDR, VNS and BBR respectively). Accordingly, the aircraft initially
climbed to the base/ceiling altitude, stabilized and made horizontal flight along the projected track
for about 30 min before climbing up/ down to the next higher/ lower levels and stabilizing. This
procedure was repeated for all levels (~ 0.5, 1, 1.5, 2, 2.5 and 3 km a.g.l.) until the last level. The
ceiling altitude was restricted to 3.5 km based on the unpressurised mode of operation of the aircraft.
All the flights were carried out around mid-day since thorough vertical mixing is established by the
daytime convective boundary layer eddies.
2.2. On-board Instrumentation
The measurements were carried out aboard the instrumented aircraft (Beechcraft-200) fitted with an
iso-kinetic inlet, mounted (front facing) at the bottom of the fuselage for aspirating ambient aerosols
and detailed in earlier papers (Babu et al., 2016; Vaishya et al., 2018; Gogoi et al., 2019). A constant
volumetric flow of 70 LPM was maintained using an external pump connected to the main inlet
assembly, which provided iso-kinetic flow for the average speed of 300 km/hr maintained by the
aircraft during the entire campaign. The efficiency of this inlet system has been already proven in
several previous campaigns (Babu et al., 2016; Nair et al., 2016; Gogoi et al., 2019).
*Measurement of aerosol size distribution*
A factory-calibrated, Aerodynamic Particle Sizer (APS) spectrometer (TSI, Model: 3321) is used
for the measurement of aerosol size distribution. It measures size-resolved number concentration of
the ambient aerosols in the size range from 0.5 to 20 μm, over 52 channels spaced equally in
logarithmic size bins; at a sampling frequency of 1-minute. Aerosol particles in this size range is
most important in influencing the optical (scattering and extinction) and CCN and ice nuclei (IN)
characteristics.
The APS measures the concentration of particles in terms of their aerodynamic diameters by
comparing the velocity of particle (controlled by an accelerating flow field) to that of a unit density
sphere having same velocity. Particle velocity is estimated from the measurement of time of flight
(Mitchell and Nagel, 1999). In the present study a sheath flow at 4 LPM (litres per minute) was
maintained against the sample flow of 1 LPM. The instrument automatically adjusts the flow rates
with changes in ambient pressure to maintain the specified flow rates. Occasionally, when the
aircraft passes through clouds, the aerosol number concentration shot up from the otherwise stable
values. Such outliers are removed following 2σ criteria, wherein data points at a particular level
lying outside 2σ values of the level-average were removed. The number of such screened out points
were < 3% of the total. The consistency in the flow was periodically checked each time, before start
of measurements from the new base station. Similarly, the optical components and tubing of the
system were cleaned immediately after moving to a new base station.
The TSI-APS (3321) is suitable for operating at 10 to 90% RH (non-condensing) and 10 - 40 °C
ambient temperature. For BBR, it is likely that aerosols grew under high RH conditions but might
have also shrunk due to higher instrument temperature as compared to ambient. However, more
controlled laboratory experiments are required to ascertain the response of the APS to hygroscopic
growth of particles.

*Measurement of Black Carbon aerosols*
Mass concentration of ambient BC aerosols was estimated using a 7-channel aethalometer (Model:
AE-33, Magee Scientific, USA), which measures the attenuation of light that passes through the
aerosol laden filter at wavelengths 370, 470, 520, 590, 660, 880, and 950 nm. The loading (or
shadowing) effect arising out of the successive deposition of aerosols in the filter media is
automatically compensated in real-time in the new-generation Aethalometer; while the multiple
scattering effects were minimised by using advanced filter tape material (Drinovec et al.,2015). In
the present study, BC mass concentrations were obtained at 1-minute interval by operating the
aethalometer at 50% of the maximum attenuation, and a standard mass flow rate of 2 LPM under
standard temperature ($T_O$, 293 K) and pressure ($P_O$, 1013 hPa). As the unpressurised aircraft climbed
higher, the instrument experienced ambient pressure (P) and temperature (T). In order to maintain
the set mass flow, the pumping speed of the instrument was automatically increased (through internal
program) to aspire more volume of air. However, the volume of air aspirated at ambient pressure
and temperature requires to be corrected to standard atmospheric condition for the actual estimate
of BC (Moorthy et al., 2004). Thus, the actual volume of air aspirated by the Aethalometer at
different atmospheric level is,
$$V = V_o \frac{P_o T}{P T_o}$$

Thus, true BC mass concentration ($M_{BC}$) is
$$M_{BC} = M_{BC}^* \left[ \frac{P_o T}{P T_O} \right]^{-1} \tag{1}$$

Here, $M_{BC}^*$ is the instrument measured raw mass concentration of BC at ambient pressure and
temperature. Details of the aethalometer principle, operation, uncertainty involved and error budget
are reported in several earlier literatures (Weingartner et al., 2003; Arnott et al., 2005; Gogoi et al.,
2017). In general, the instrumental uncertainty ranges from 50% at 0.05 µg m$^{-3}$ to 6% at 1µg m$^{-3}$
(Corrigan et al., 2006) and the uncertainty in the estimation of absorption coefficients is around 10%
(Vaishya et al., 2018).
2.4. General synoptic meteorology during the campaign
The meteorological conditions across the IGP during the campaign period was generally hot (surface
temperature, T ~ 34.7 ± 2.8 ºC at JDR, 39 ± 1.9 ºC at VNS and 32.8 ± 3.6 ºC at BBR at the time of
flight take off), with low to moderate relative humidity (RH) at JDR(RH ~ 40%) and VNS (RH ~
60%). The values of RH at BBR was relatively higher (as high as 80%) associated with its coastal
proximity, in addition to the influence of mild pre-monsoon rainfall during the first (01-June-2016;
light rain during noon), third (03-June-2016; heavy rain ~ 60 mm in the night) and fourth (04-June-
2016; light rain in the morning and during noon) days of observations. The records of T and RH
were obtained from the sensors on-board the aircraft, while the rainfall data was obtained from the
airport meteorological department at BBR.
2.3. Supplementary data
Supplementary data used in this study include aerosol backscattering coefficients and depolarization
ratio measured by the *Cloud Aerosol Transportation System* (CATS) aboard the International Space
Station (ISS). The CATSis a comprises of an elastic backscatter lidar consisting of two high
repetition rate (4-5 kHz), low energy (1-2 mJ) Nd:YVO$_4$ lasers operating at three wavelengths (1064,
532, and 355 nm). The receiver subsystem consists of a 60 cm telescope having a 110 micro-radian
field of view, photon-counting detectors, and associated control electronics (Yorks et al., 2014;
2016).As the altitude of ISS orbit is about 405 km (51-degree inclination), CATS provides a
comprehensive coverage of the tropics and mid-latitudes, with nearly a three-day repeat cycle. Level
2 data of CATS (https://cats.gsfc.nasa.gov/data/) are used (Lee et al., 2018) in the present study,
which provides the geophysical parameters, such as the vertical feature mask, profiles of cloud and
aerosol properties (i.e. extinction, particle backscatter), and layer-integrated parameters (i.e. lidar
ratio, optical depth). In addition, types of aerosols are also derived based on CATS typing algorithms
where eight aerosol types (in CATS mode 7.1) are identified: volcanic, dust, dust mixture,
clean/background, polluted marine, marine, polluted continental and smoke. Incorporating the
information of backscatter color ratio (1064/532-nm) and spectral depolarization (ratio of
perpendicular to parallel backscatter) ratio(1064/532-nm), Mode 7.1 provides the characteristic of
aerosol regimes (York et al., 2016) as below:

**Table-1**: Classification of aerosol types for CATS mode 7.1 (York et al., 2016).

| Aerosol Type | Aerosol feature base | Depolarization ratio $(\delta'_{1064})$ | Color Ratio $(\gamma'_{1064})$ |
|---|---|---|---|
| Volcanic | > 10 km | - | - |
| Dust | < 10 km | > 0.3 | - |
| Dust mixture | < 10 km | $0.2 > \delta > 0.3$ | - |
| Clean/background | < 10 km | - | $< 0.0005\ \text{sr}^{-1}$ |
| Polluted marine | < 10 km | $\delta'_{1064}/\,\delta'_{532} > 50\%$ | $\gamma'_{532}/\,\gamma'_{1064} < 1.75$ |
| Marine | < 10 km | $\delta'_{1064}/\,\delta'_{532} < 50\%$ | $\gamma'_{532}/\,\gamma'_{1064} < 1.75$ |
| Polluted continental | < 10 km | $\delta'_{1064}/\,\delta'_{532} > 50\%$ | $\gamma'_{532}/\,\gamma'_{1064} > 1.75$ |
| Smoke | < 10 km | $\delta'_{1064}/\,\delta'_{532} < 50\%$ | $\gamma'_{532}/\,\gamma'_{1064} > 1.75$ |



**3. Results and discussion**
3.1 Aerosol number size distributions
Aerosol number size distributions [dN/d (logDp)], representative of each of the 3 sub-regions of
IGP, are presented in Figure 3; the panels from left to right representing the sub-regions JDR, VNS
and BBR, from the west to east IGP.

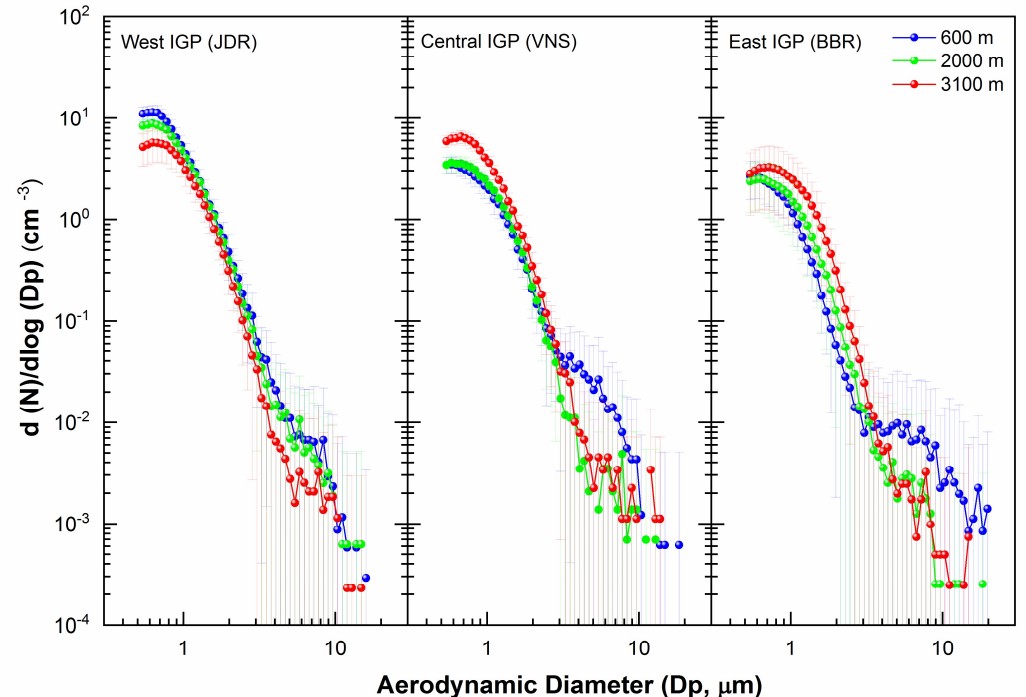


**Figure-3:** Aerosol number size distributions (mean profiles averaged for all the days) at three
distinct altitudes of JDR, VNS and BBR, representative of (i) near the surface (600 m above ground
level) having proximity to emission sources, (ii) in the upper ABL (2000 m above ground level) and
(iii) in the free troposphere (3100 m). Vertical bars over the points are the ensemble standard
deviations. Individual size distributions at different heights of ~ 500 m interval are given in
supplementary figure-S1.
Three distributions are shown for each station, representative of (i) near the surface with proximity
to emission sources (600 m AGL), (ii) in the upper ABL (2000 m AGL) and (iii) in the free
troposphere (3100 m AGL) following the mean ABL heights (1.3 ± 0.5 km, 2.3 ±0.5 km and 1.4 ±
0.2 km for JDR, VNS, and BBR respectively; Vaishya et al., 2018) at local noon time. Aerosol
number concentration below 0.542 μm are not size-classified and represented as a single count
(between 0.3 and 0.542 μm) are shown as a function of altitude in Figure 4 (a).

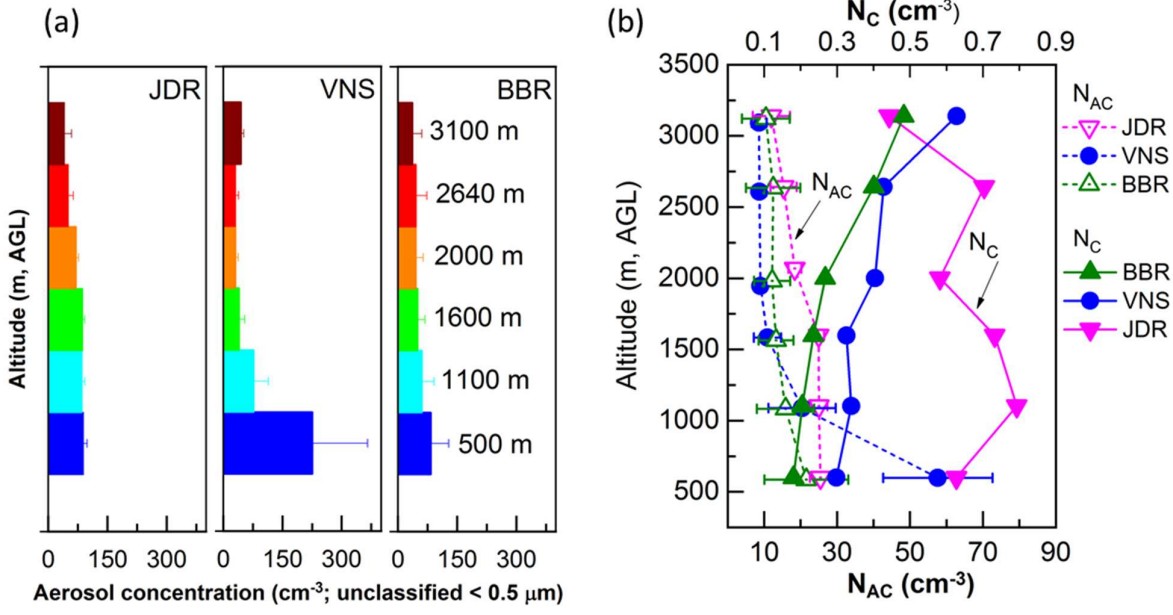


**Figure-4:** Vertical profiles of aerosol number concentrations;(a) between 0.3 and 0.54 μm (in the
unclassified size range of APS); (b) in the accumulation and coarse mode size range (between 0.3
and 20 μm, denoted by $N_{AC}$) along with coarse mode number concentrations ($N_C$).
The figures clearly reveal that at all altitudes above different stations, the size distributions are
consistently bimodal, with a prominent accumulation mode (<1μm) and a weaker secondary mode
(>1μm).The concentration of particles in the unclassified size regime (below 0.542 μm), showed a
gradual decrease with increase in altitude at all stations and a spatial distinctiveness with highest

near surface concentration in the Central IGP (most anthropogenically impacted sub-region of the IGP) depicting sharper altitude variation as against the other two sub-regions.

As it is well-established that during pre-monsoon/ prior to the onset of monsoon, both the natural and anthropogenic aerosol species coexist in large abundance over the IGP, we examined in Figure 4b, the altitude profiles of accumulation mode aerosols (concentration below 1 μm), which are mostly attributed to be of anthropogenic origin and coarse mode aerosols (above 1 μm), which are mostly of natural origin. Accumulation mode aerosol concentration showed only weak altitudinal dependence above 1 km at all the sub-regions, though at VNS, there was a sharp increase in the concentration below 1 km, obviously due to source-proximity. This feature is seen in Figure 4a also. This observation is supported by the collocated measurements of aerosol total number concentrations ($N_T$) as measured by a condensation nuclei (CN) counter aboard the aircraft (Jayachandran et al., 2020) in the size range above 2.5 nm, showing highest values of $N_T$ in the entire altitude range of measurements over VNS. On the other hand, the vertical profiles of coarse mode aerosol concentrations ($N_C$) showed significantly large abundance over the western IGP (arid/ semi-arid regions) represented by JDR, similar to the spring time observations reported by Gogoi et al., (2019).

These observations are also in-line with the reported values of dust fractions (Vaishya et al., 2018) during the same campaign, showing the enhancement of dust fraction from 10 to 20 % at 300 m to more than 90 % above 2 km altitude at JDR; while smallest dust faction (< 10%) was observed at BBR in the entire altitude range. Over the central IGP, synoptic wind-driven desert dust aerosols, leads to elevated layers of aerosols having higher dust fraction (>50%). However, it should be noted that dust over the central IGP is more absorbing in nature because of its mixing with other anthropogenic emissions (such as BC; Vaishya et al., 2018), while that over western IGP is rather pristine in nature. Thus, quantification of the absolute magnitude of coarse mode aerosol concentrations is very important to understand the significance of elevated aerosol load on radiative perturbations.

The increasing concentration of coarse mode particles with the increase in altitude across the entire IGP is another interesting feature in the present study; which is most conspicuous at the central IGP and least at the west, implying their increasing role at higher altitude; probably due to the lofted regional dust and advected mineral dust from west Asian regions.

With a view to quantifying this, the size distribution spectra are averaged for each altitude level and
for each station. From these spectra, the geometrical mean diameter (Dg) is estimated as a function
of altitude, using the following equation

298           $D_g = exp\left[\frac{\sum_l^u n_i \, ln(D_{pi})}{N}\right]$                                    (2)

where $D_{pi}\left(= \sqrt{(D_i * D_{i+1})}\right)$ denotes the geometric midpoint of each channel of the APS, $n_i$ is the
particle concentration in $i^{th}$ channel and $N = \sum_l^u n_i$ is the total concentration. Accordingly, Dg of a
spectrum of particles is the 50% probability point of an equivalent diameter having half of the
particle concentrations larger than this size and remaining half is below that. The vertical profiles of
Dg and mode $\left(= D_p(n_{max})\right)$ of the distributions are shown in Figure 5. It clearly shows the increase
of the coarse mode fraction in the size distribution; with both the mode and Dg showing a steady
increase with altitude; especially Dg. The rate of increase of Dg with altitude increases from west to
east across the IGP, with highest values at BBR (Figure 5b). In the central IGP where mixed aerosol
type prevails, the increase in Dg within the ABL is rather weak, but in the free troposphere it
increases more sharply probably due to the faster decrease in the accumulation mode concentration
(Figure 4) or the prevalence of advected dust at higher altitudes or both.
The observations that have foregone reveal the non-uniform distribution of dust and anthropogenic
sources of aerosols. Nearly steady values of $N_C$ in the entire column at JDR are attributed to the
strong convective mixing of coarse mode dust aerosols up to the lower free tropospheric region. On
the other hand, altitude variation of accumulation and coarse mode aerosols are relatively more
fluctuating at BBR and VNS, compared to that at JDR (Figure 4b) as indicated by the profiles of Dg.

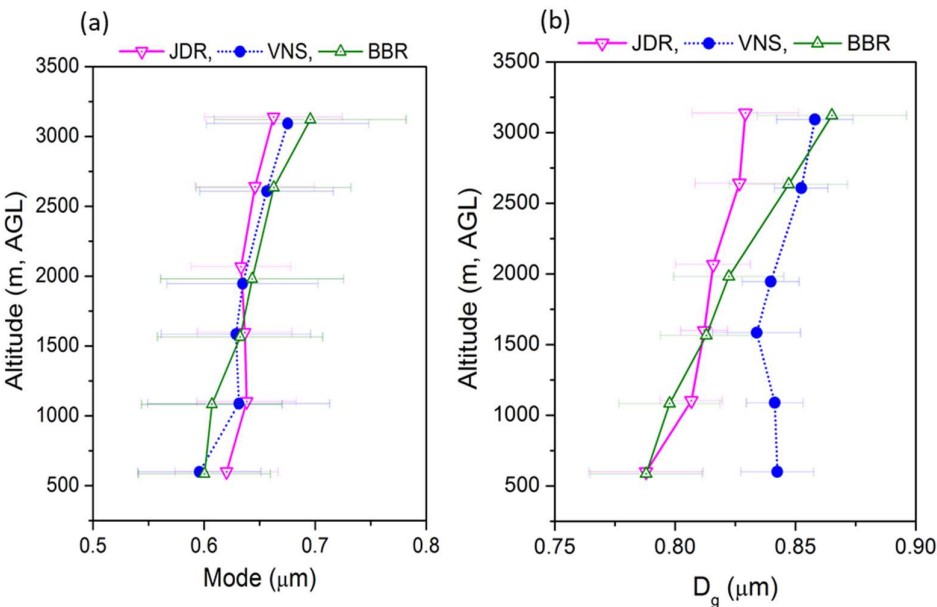

**Figure-5:** Vertical profiles of (a) mode and (b) geometric mean diameters (Dg) of aerosol number size distributions at different heights above the ground level, indicating the change in the pattern of distribution with altitude and from the western to the eastern part of the IGP.

Apart from the number-weighted expression of aerosol size distributions, the mass-weighted distributions carry useful information for quantifying regional distinctiveness of the dominance of coarse mode particles. Even though the fine mode aerosols are extremely numerous in the atmosphere and important for microphysical processes, they represent only a very small proportion of total particle mass; whereas coarse mode particles, even though far less numerous, have significant mass/ volume. In simple terms, particle number concentrations are dominant in the fine mode ($< 0.1$ μm), the surface area is predominantly in the accumulation mode (0.1 to 1 μm), and the volume, and hence mass, is divided between the accumulation mode and coarse particle mode. In the present study, since the size range of particle counts are confined in the accumulation and coarse mode regimes (between the 0.5 and 20μm), quantitative picture of aerosol mass concentrations is obtained by assuming a uniform density equal to 2 g cm$^{-3}$ following Moorthy et al., (1998) and Pillai et al., (2001). Since the size-resolved particle densities are not known, we did not use effective density (mass-mobility relationship defined as the mass of the particle divided by its mobility equivalent volume) of particles to calculate the mean particle mass size distributions.

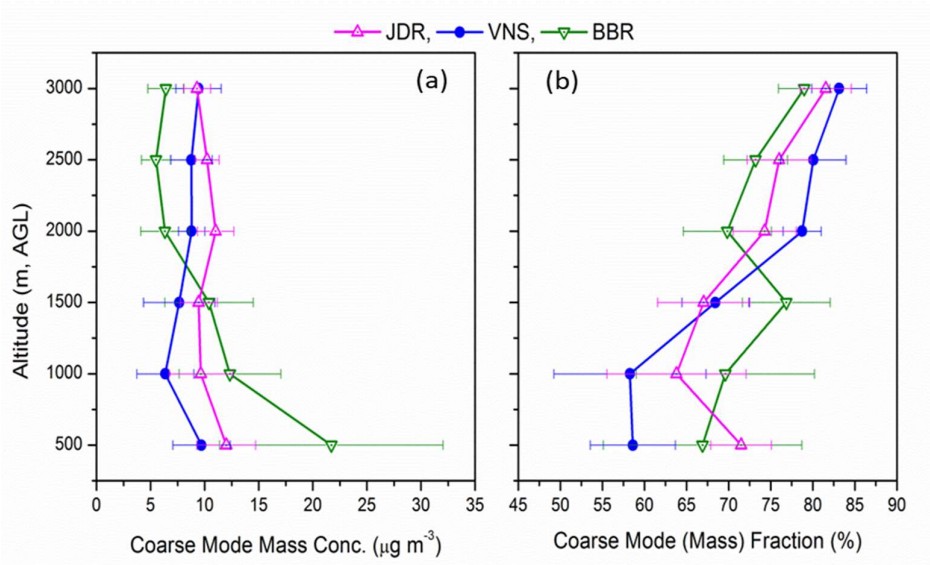

**Figure-6: (a)** Vertical profiles (mean and standard deviations) of coarse mode aerosol mass concentrations ($M_C$). The values are derived from the aerosol number concentrations at different size bins, assuming a density of 2 gm/cm$^3$;(b) Vertical profiles of aerosol coarse mode fractions ($F_{MC}$) at different locations.

Figure 6a shows the altitudinal variation of coarse mode aerosol mass concentrations over all the observational sites, along with the values of coarse mode mass fractions ($F_{MC}$). Over VNS and JDR, consistently higher values of $M_C$ were seen in the entire altitude range. This is in line with the higher values of coarse mode aerosol concentrations ($N_C$) at these sites, JDR being the highest. On the other hand, the values of $M_C$ at BBR decreased significantly from the surface to lower free-tropospheric region. The higher values $M_C$ observed near the surface at BBR can be attributed to the influence of local sea-salt aerosols; however not affecting the values of Dg, due to the abundance of accumulation mode aerosols over this site.

Similar to that of $N_C$, $F_{MC}$ showed (Figure-6b) gradually increasing values with altitude at all the locations. The high values of coarse mode mass fraction and an increasing trend with altitude is indicative of the role of upper level transport of dust from the western desert regions, in addition to those contributed locally due to thermal convective processes. As compared to other two stations, highest value of $F_{MC}$ (~ 70%) near the surface was seen at JDR indicating the role of arid nature of the region. This exercise clearly explains the abundance coarse mode dust decreasing from west to east; along with an increase in the contribution of anthropogenic fine/ accumulation mode aerosols.

With a view to examine the transport of mineral dust (by the synoptic winds), the spatial distributions of UV-aerosol index, aerosol types and aerosol absorption optical depth (AAOD); all derived from the Level-3 OMAERUVd data product (daily, 1.0 degree x 1.0 degree) from Ozone Monitoring Instrument (OMI, on-board Aura satellite; Levelt et al., 2006), are examined. OMAERUV uses the pixel level Level-2 Aerosol data product of OMI at three wavelengths (355 nm, 388 nm and 500 nm) to derive AAOD. Higher values of AAOD at 388 nm are indicative of the presence of dust or biomass burning aerosols. This is because absorption by dust and organic carbon from biomass burning sources have strong wavelength dependency, with higher absorption at near-UV wavelengths. As the period of this campaign was devoid of major fire activities over the study region (northern India) which normally peaks in April to May and October to November, corresponding to burning after the wheat and rice harvests (Vadrevu et al., 2011; Venkataraman et al., 2006), the AAOD values would be representative of dust loading. This aspect is conformed in the subsequent section using lidar depolarization ratio.

Figure 7a-d shows the spatial distributions of UV aerosol index, aerosol type and AAOD at 388 nm and 500 nm, while the synoptic winds are shown in Figure 7e. A very good association between the westerly advection and dust loading extending from west to central IGP is noticeable from the figure. This lends further support to the role of advected dust leading to higher $M_C$ and $F_{MC}$ at higher altitudes, seen in figures 6. In this context, it is also worth noticing that based on observational data and regional climate modeling, Banerjee et al., (2019) have clearly shown (in their Figure 7) the significant vertical extent of dust loading, both of local and remote origin, during pre-monsoon and summer across the IGP reaching altitudes as high as 600 hPa.

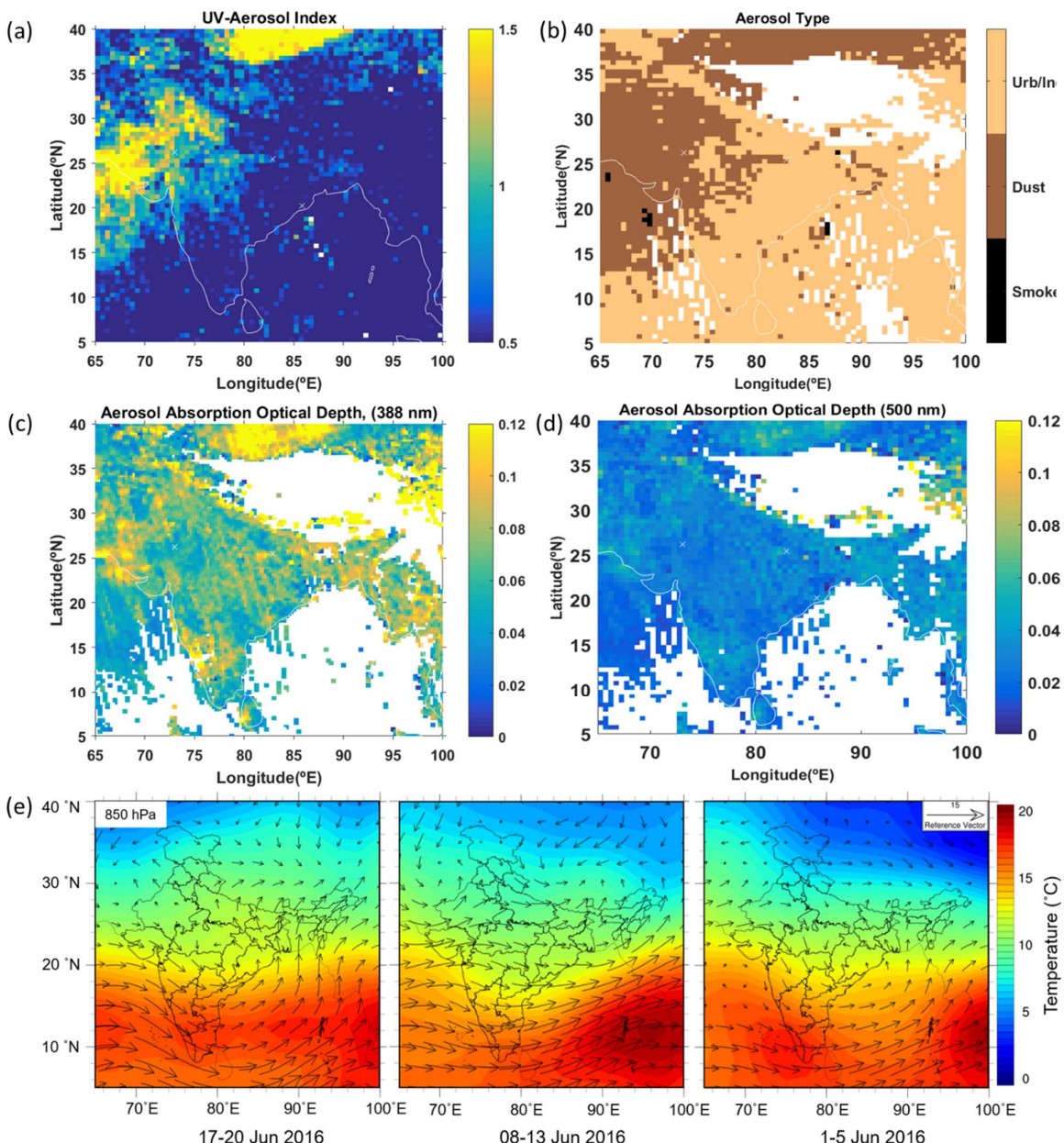

374

**Figure-7:** Spatial distribution of (a) UV aerosol index, (b) aerosol type, (c) aerosol absorption optical depth (AAOD) at 388 nm and (d) AAOD at 500 nm during June 2016. (e) Synoptic wind and temperature at 850 hPa.

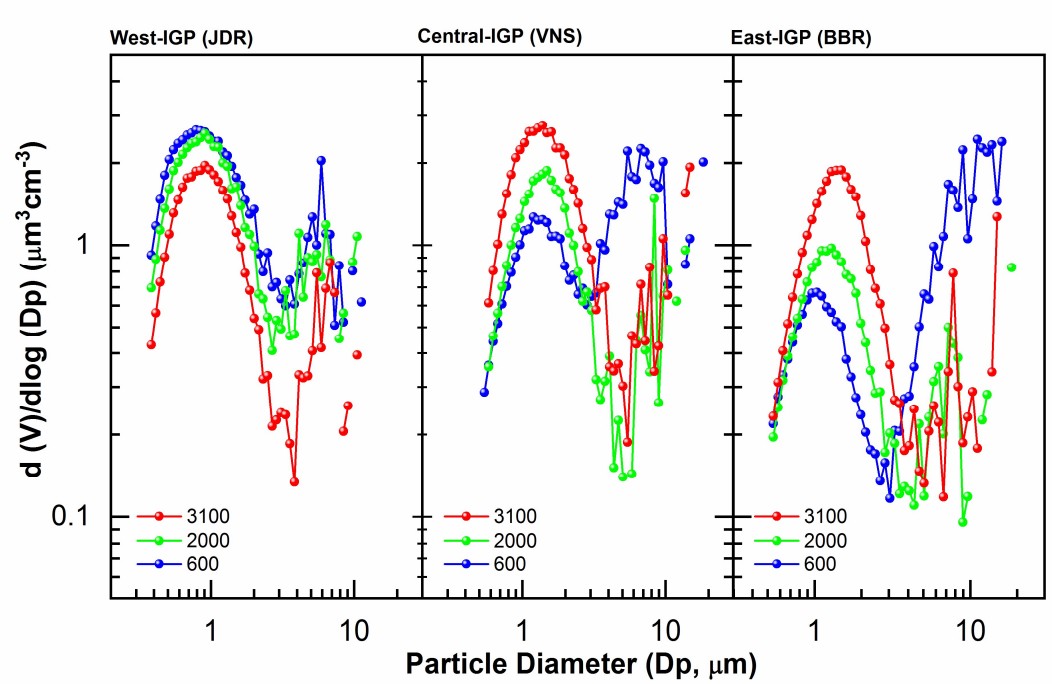

**Figure-8:** Aerosol volume size distributions (mean profiles averaged for all the days) at three distinct altitudes (600 m, 2000 m and 3000 m) of the atmosphere (shown by different color) over JDR, VNS and BBR.

The volume size distribution of aerosols (shown in Figure 8) at three distinct altitude regions of the atmosphere also clearly shows the altitudinal change in the pattern of distribution, changing from coarse mode dominance near the surface to accumulation mode dominance at the ceiling altitude over BBR. While those at JDR, the pattern of distributions remains same in the entire column. Similar to JDR, VNS also depicted significant enhancement in coarse mode aerosols in the upper levels (at 2 and 3 km altitudes) of the atmosphere. Similar to these observations, based on the collocated spectral scattering properties of aerosols obtained during the same experiment, Vaishya et al., (2018) have reported that, the aerosol population changes from super-micron mode dominant natural aerosols to sub-micron mode dominant anthropogenic aerosols, as we move from west to east in the IGP. Moreover, the large abundance of coarse particles (>2µm) along with significant fine/ accumulation mode aerosols in the column highlights the complex mixture of dust with other anthropogenic components in all the three regions, making a complex scenario for aerosol radiation and aerosol cloud interaction processes. Based on the combination of satellite remote sensing and regional climate model simulations, Banerjee et al., (2019) have also shown the presence of dry elevated layer of dust (at altitudes between 850 and 700 hPa; taking place in multiple layers) during

June across the IGP, transported from the Thar Desert to the northern Bay-of-Bengal. To ascertain
this further, we have examined the data from CATS aboard ISS.
3.2 Inferences from the CATS data
Geophysical parameters derived from the CATS on-board ISS are very useful to infer on aerosol
features in the atmospheric column, especially at altitudes above the ceiling altitude of the aircraft
(3.1 km). In the present study, we have considered three products from CATS for the campaign
period, viz. (i) depolarization ratio, (ii) attenuated backscatter coefficients and (iii) aerosol types.

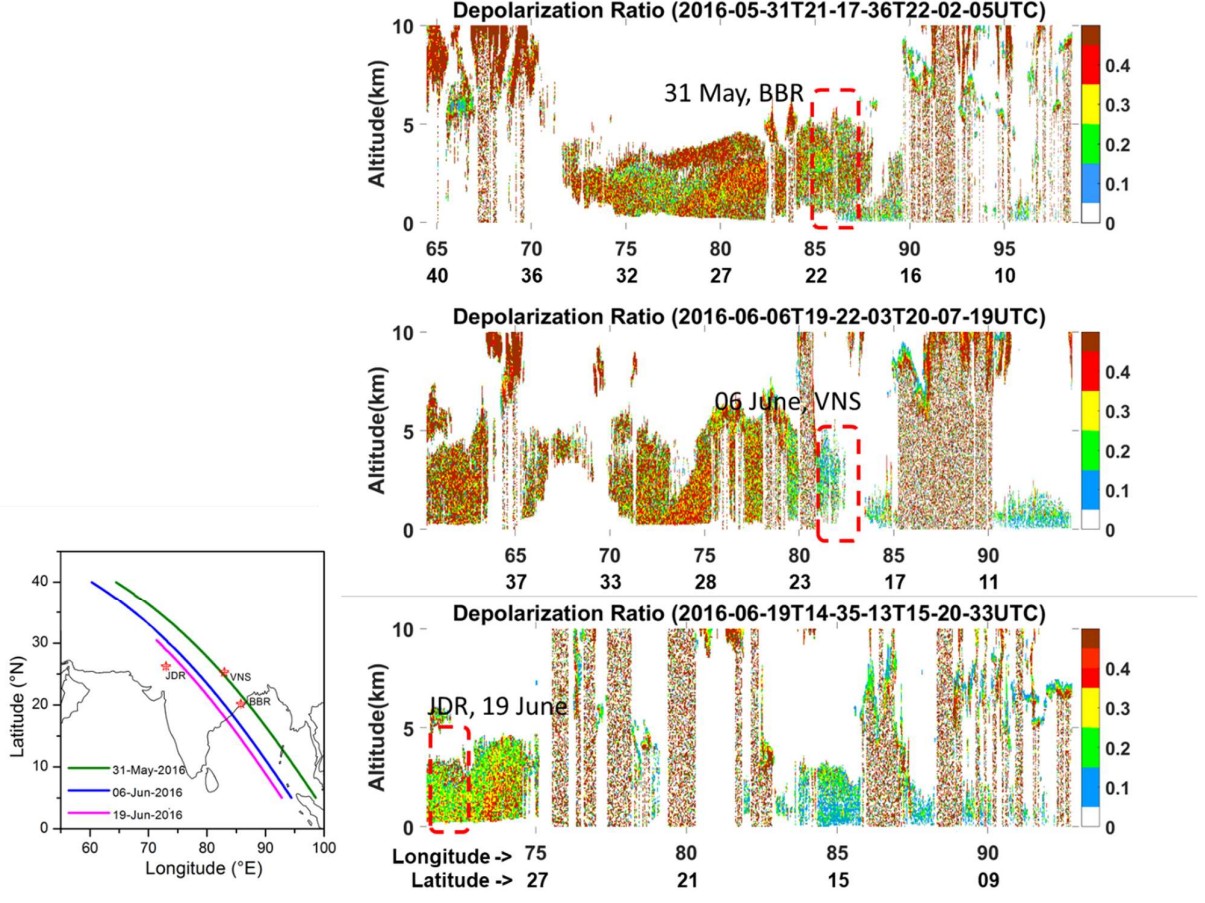


**Figure-9:** Aerosol Depolarization Ratio [obtained from Cloud Aerosol Transportation system in
International Space Station (ISS)] for three different passes of the ISS over the three sub-regions
during the period of aircraft observation. The tacks of the CATS are shown by the solid lines in the
left panel and the rectangular boxes in the right panels show the data over the sub-regions.

Figure 9 shows the vertical cross-section of depolarization ratio for three passes during the campaign period and close to the three sub-regions (identified by the rectangular boxes in the figure). Higher values (~0.3) of depolarization ratios are seen in the western IGP (JDR, bottom panel), suggesting the dominance of non-spherical (dust) particles. The depolarization ratio decreases towards east across the IGP, with values equal to 0.1 at the central IGP, and ~ 0.2 in the eastern site BBR. These lend additional support to the inference on the influence of dust aerosols during the campaign period. Supporting the patterns of depolarization ratio, aerosol types (from CATS mode 7.1) in Figure 10a indicates significant presence of dust at JDR, while the aerosol types over VNS and BBR are mixture of dust, polluted continental and carbonaceous aerosols. Vertical profiles of total attenuated backscatter coefficients show the vertical extent of the aerosol layer to be as high as 5 km (as has been shown by Banerjee et al 2019) over all the sites (Figure 10b).

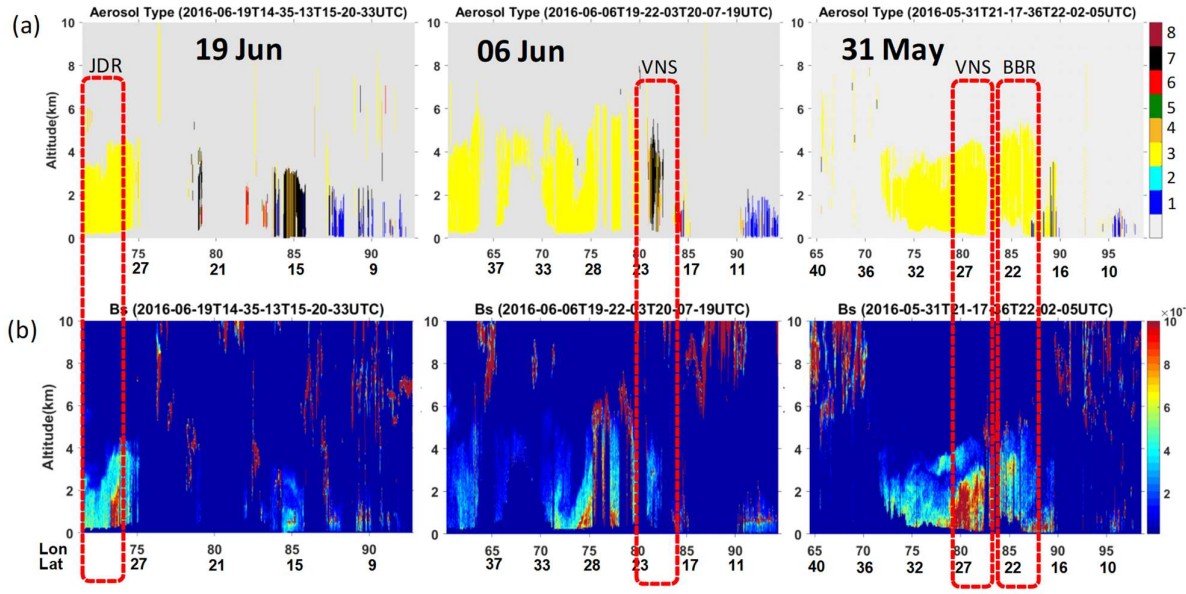

**Figure-10**: Transects of (a) Aerosol types (1- Marine, 2- Marine Mixture, 3- Dust, 4- Dust Mixture, 5- Clean/ Background, 6- Polluted Continental, 7- Smoke, 8- Volcanic), and  (b) Backscatter coefficients  (Bs, $km^{-1}Sr^{-1}$) at 1064 nm obtained during the period of aircraft observation corresponding to the overpass of the ISS.

3.3 Vertical profiles of BC

BC is the chief anthropogenic absorbing aerosol species, and the IGP is known to be among the global hotspots (Govardhan et al., 2019). The height resolved information on $F_{BC}$ is important not

only in radiative forcing, but on CCN activation as well (Bhattu et al., 2016). Collocated
measurements of BC during SWAAMI - RAWEX have been used to examine the vertical profiles
of BC and its variation across the IGP prior to onset of the Indian summer monsoon. Figure 11a
shows the vertical profiles of BC for the three sub-regions. Each profile is the average of all the
profiles obtained from measurements made from each of the base station. It is seen that, BC remained
low (~1 $\mu g\ m^{-3}$) and depicted very weak altitude variations at the western and eastern IGP regions,
while in the central IGP there is a rapid decrease of BC from the high value (~ 3$\mu g\ m^{-3}$) near the
surface. Above 2 km, all the profiles overlap though a weak increase is indicated over BBR, which
is examined later. The very high values of BC close to the surface at VNS are attributed to the wide-
spread anthropogenic activities in the Central IGP including the cluster of thermal power plants in
that region. Consequently, the columnar concentration of BC (integrated up to 3 .1 km) is also the
highest at VNS.

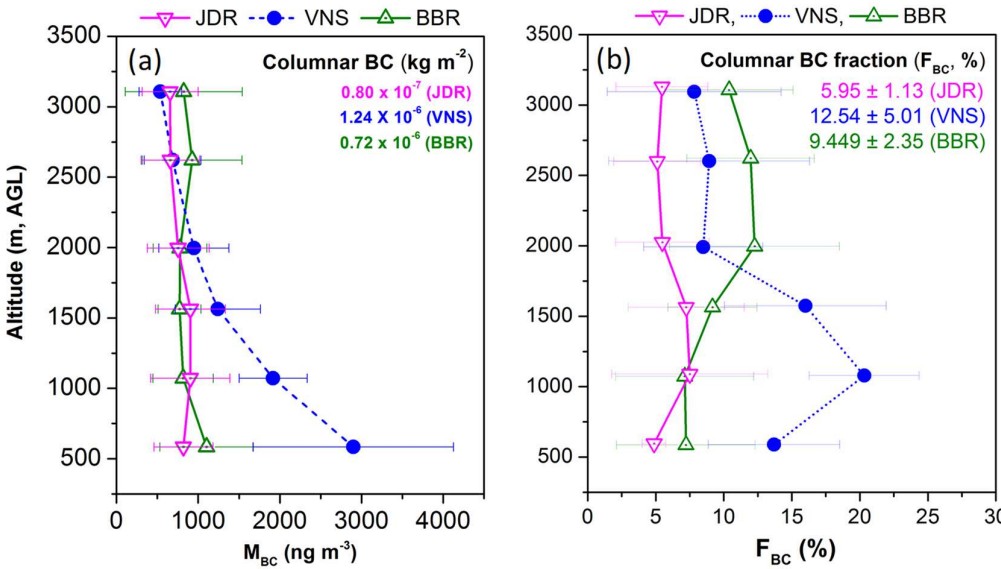

**Figure-11:** Vertical profiles of (a) mean values of BC mass concentrations ($M_{BC}$) and (b) BC mass
fractions ($F_{BC}$) at JDR, VNS and BBR.
However, the vertical profiles of the fractional contribution of BC ($F_{BC}$) to the total composite
aerosol mass (estimated from the volume size distribution, considering a uniform density of 2 gm/cc,
especially in view of the abundance of dust) shows (Figure11b) sub-regional distinctiveness. It
remains the lowest (~6%) in the western IGP, with very little altitude variation. In the central IGP,
$F_{BC}$ is quite high (~15 % to 20%) within the ABL and drops off fast above 2 km approaching the
values seen for the western IGP. $F_{BC}$ depicts an elevated peak at around 1 km above ground level at
VNS, while at BBR, higher $F_{BC}$ values occur at still higher altitudes at BBR, where the near-surface
values are much lower and comparable to those at JDR. There is a steady increase in $F_{BC}$ from near
surface to higher altitudes, and above 2 km, the values are comparable to the peak values seen at
VNS (at ~ 1 km altitude). Despite this, the integrated BC concentration comes in between those of
JDR and VNS, mainly because of the large values occurring in the lower atmosphere at VNS. It may
be recalled that based on SWAAMI - RAWEX aircraft measurements, Vaishya et al., (2018) have
reported that while the scattering characteristics remained uniform across the IGP, the absorption
coefficients showed sub-regional distinctiveness leading to a west to east gradient (decrease) in the
vertical structure of single scattering albedo (SSA).
Investigation of the vertical profiles of BC mass concentrations on individual days (Supplementary
Figure-S2) helps to see the distinctiveness at each sub-region, resulting from the spatially
heterogeneous nature of emission sources and advection, especially at BBR where the inland
profiles, made during sorties perpendicular to the coastline (on 2nd and 3rd June) show significantly
higher values of BC at higher altitudes than those along the coastline. At BBR, this arises mainly
because of spatially heterogeneous source impacts. The regions towards the northwest of BBR are
characterized by large scale urban and industrial activities (Ambient air quality status and trends in
Odisha: 2006 - 2014). Similarly, near surface BC concentrations at VNS was higher when the flight
sorties were confined to NE, NW and SW of the city Centre, while the values in the SE sector was
lower. On the other hand, at JDR, the profiles revealed a better spatial homogeneity.
To quantify the climatic implications of BC, the heating rate profiles of BC is examined based on
the estimation of shortwave direct radiative forcing (DRF) due to BC alone. The DRF due to BC
represents the difference between the DRF for aerosols with and without the BC component. The in-
situ values of scattering ($\sigma_{sca}$) and absorption ($\sigma_{abs}$) coefficients measured on-board the aircraft were
used to estimate spectral values of AOD (layer-integrated $\sigma_{sca} + \sigma_{abs}$), single scattering albedo (SSA)
and asymmetry parameter (g) for each level, assuming a well-mixed layer of 200 m above and below
the measurement altitude (details are available in Vaishya et al., 2018). The layer mean values of
AOD, SSA and Legendre moments of the aerosol phase function (derived from Henyey–Greenstein
approximation) are used as input in the Santa Barbara DISORT Atmospheric Radiative Transfer
(SBDART, Ricchiazzi et al., 1998) model to estimate diurnally averaged DRF (net flux with and
without aerosols) at the top ($DRF_{TOA}$) and bottom ($DRF_{SUR}$) of each of the layers. The atmospheric
forcing ($DRF_{ATM}$) for each of the levels is then estimated as
$DRF_{ATM} = DRF_{TOA} - DRF_{SUR}$                              (3)
In order to estimate the forcing due to BC alone, optical parameters for aerosols were deduced again.
For this, values of $\sigma_{abs}$ were segregated to the contributions by BC ($\sigma_{BC}$) and OC ($\sigma_{OC}$), where $\sigma_{BC}$
were estimated following inverse wavelength dependence of BC (e.g., Vaishya et al., 2017).  Based
on this, a new set of AOD and SSA for BC-free atmosphere is calculated and fed into SBDART for
estimating $DRF_{ALL-BC}$ without the BC component. Thus, DRF due to BC is
$$DRF_{BC} = DRF_{ALL} - DRF_{ALL-BC} \tag{4}$$
Here, $DRF_{ALL}$ represents forcing due to all the aerosol components, including BC.

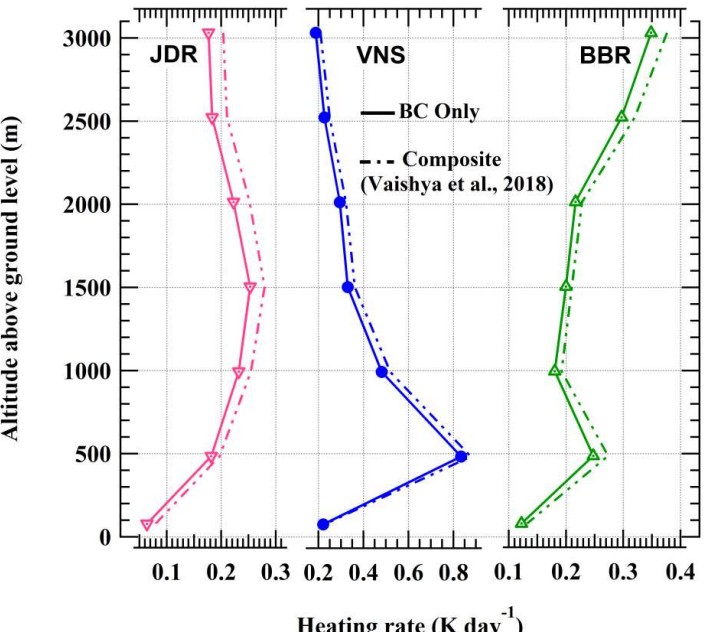


Figure-12: Vertical profiles of atmospheric heating rate due to BC (solid lines) and composite
(dashed lines) aerosols for the regions of the IGP: (a) JDR in western IGP, (b) VNS in central IGP
and (c) BBR in eastern IGP. Data for the composite heating rate profiles are from Vaishya et al.,

494    2018.

The vertical profiles of atmospheric heating rate (HR, estimated based on the atmospheric pressure
difference between top and bottom of each layer and aerosol induced forcing in that layer) due to
BC alone shows (Figure-12) maximum influence of BC in trapping the SW-radiation at VNS,
followed by BBR and JDR. Interestingly, the altitudinal profiles of heating rate are distinctly
different over the regions, BBR showing an increase with altitude, while VNS shows the opposite
pattern with maximum heating ($\sim 0.81$ K day$^{-1}$) at 500 m above ground. Enhanced heating at 500-
2000 m altitude is seen at JDR. These results indicate the dominant role of absorbing aerosols near
the surface at VNS, while the atmospheric perturbation due to elevated layers of absorbing aerosols
is conspicuous at BBR (HR ~ 0.35 K day$^{-1}$ at the ceiling altitude). The column integrated values of
atmospheric forcing due to BC alone are 7.9 Wm$^{-2}$, 14.3 Wm$^{-2}$ and 8.4 Wm$^{-2}$ at JDR, VNS and BBR
respectively.
In this context, we have examined the possible role of the large network of thermal power plants
(TPP) over the northern part of India, which is reported to have a significant contribution to regional
emissions (Singh et al., 2018). These include the emissions of $SO_2$, $NO_x$, $CO_2$, CO, VOC, suspended
particulate matter (PM2.5 and PM10, including BC and OC) and other trace metals like mercury
(Guttikanda and Jawahar, 2014; Sahu et al., 2017), dispersing over large areas through stacks. Fly
ash from coal-fired power plants cause severe environmental degradation in the nearby environments
(5-10 km) of TPP (Tiwari et al., 2019). Over the IGP, since more than 70% of the thermal power
plants are coal based, emissions of $CO_2$ and $SO_2$ hold more than 47% of the total emission share,
while the relative share of PM2.5 and NOx are ~15% and 30% (GAINS, 2012). Based on the in-situ
measurement of BC in fixed and transit areas in close proximity of seven coal-fired TPP in Singrauli
(located ~ 700 km north-west of BBR), Singh et al., (2018) have reported that BC concentration
reached as high as 200 µg m$^{-}$3 in the transit measurements. The Energy and Resources Institute,
India have also reported that emission levels of the carbonaceous (soot or BC) particles are estimated
to be around 0.061 gm/kWh per unit of electricity from Indian thermal power plants (Vipradas et al.,
2004). Based on emission pathways and ambient PM2.5 pollution over India, Venkataraman et al.,
(2018) have reported that the types of aerosols emitted from coal burning in thermal power plants
and industry in eastern and peninsular India are similar to that of residential biomass combustion.
The ongoing discussion thus clearly indicates that TPP are major sources of BC in the atmosphere.
As it is not possible to measure BC from space, to infer on the role of these emissions from thermal
power plants in causing the higher BC fraction at higher altitude over BBR, we have examined the
spatial distribution of the concentrations of the co-emitted $NO_2$ and $SO_2$ in Figure 13, in which the
locations of major coal based TPP (https://www.ntpc.co.in/en/power-generation/coal-based-power-
stations) are also marked. The data are obtained from OMI onboard AURA satellite. Higher
concentrations of $NO_2$ and $SO_2$ are readily discernible from the figure around the regions (marked
in the figure) during the period of flight experiment where there are clusters of TPP. As the energy
consumption is the highest during summer and most dependent on thermal, these TPP should be
operating to near full capacity. This provides an indirect support to the high concentrations of BC
(co-emitted) at higher levels. In general, these TPP have tall stacks (heights in the range 200 to 400
m) and aids easy ventilation to the lower free-tropospheric altitudes.

(a)
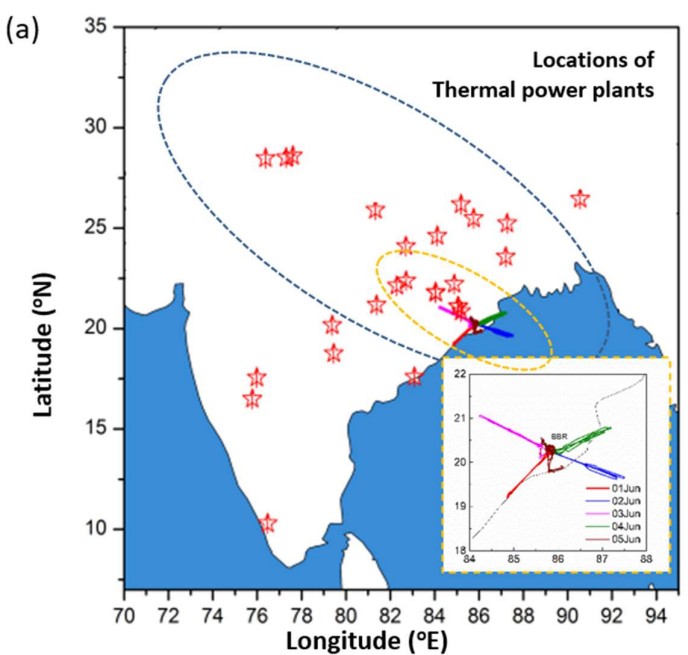

(b) **NO2 Tropospheric Column Amount (Molecules/cm2)** ×10¹⁵
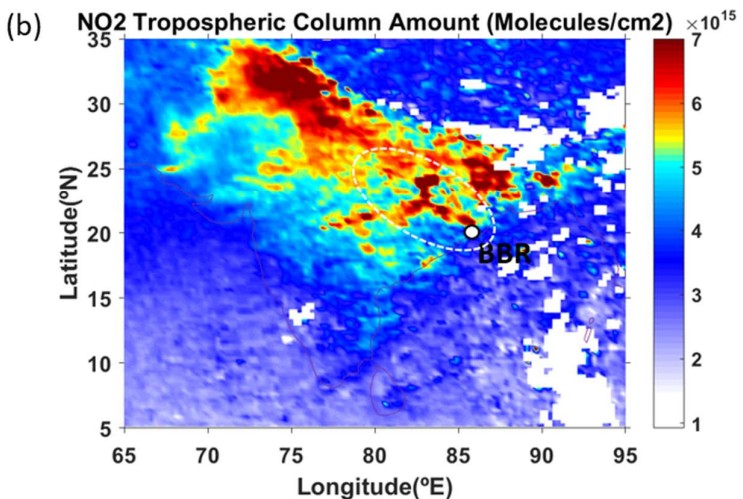

(c) **SO2 Column Amount (DU)**
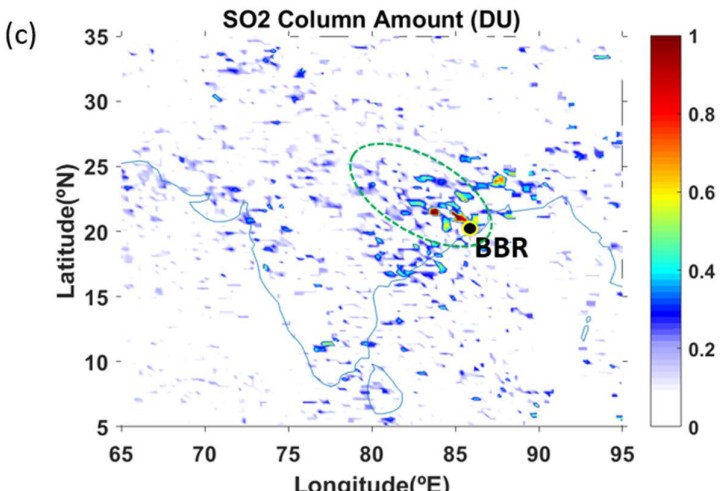


 **Figure-13**: (a) Geographic position of thermal power plants (TPP) over India (the TPP across the

IGP are bounded by the blue dashed line, and those along the flight direction of BBR are bounded
by the green dashed line), along with the spatial map of (b) $NO_2$ tropospheric column density
(molecules/cm$^2$) and (c) $SO_2$ column amount (in DU, $1DU = 2.69 \times 10^{16}$ molecules/cm$^2$) over the
northern part of India.

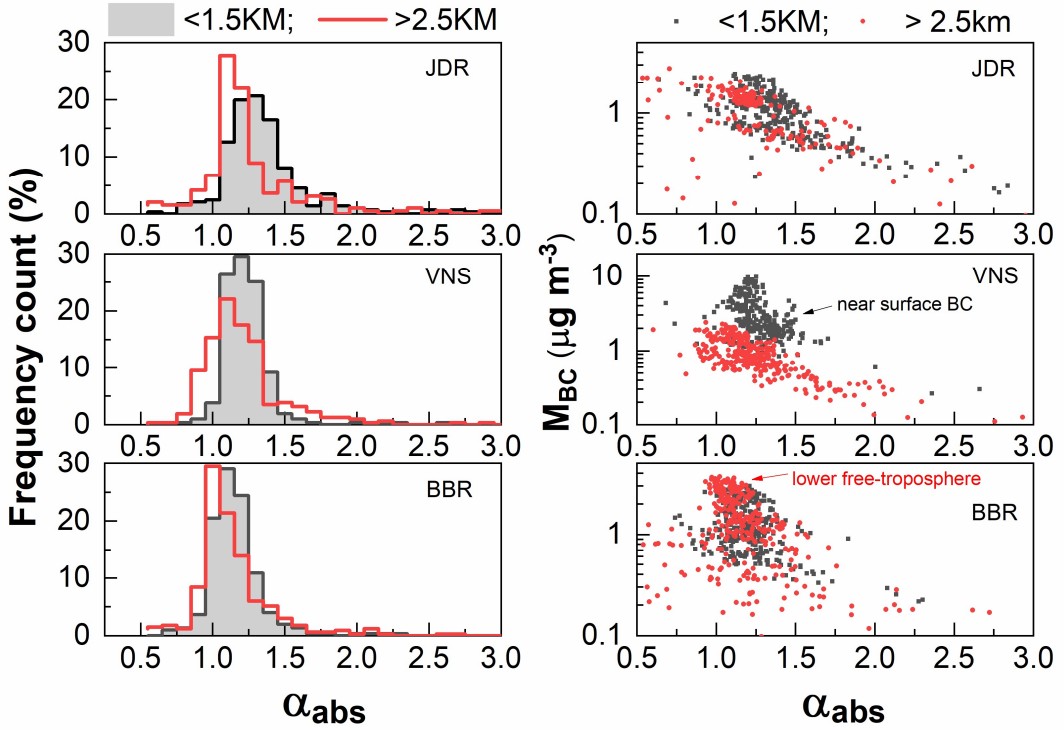


**Figure-14:** (a) Frequency of occurrences of Angstrom absorption exponent ($\alpha_{abs}$) below 1.5 km and
above 2.5 km altitude, (b) variation of BC mass concentrations corresponding to different values of
$\alpha_{abs}$ are shown in the right panels for the same two altitude regimes at distinct locations of northern
India.
To further ascertain this, the spectral properties of aerosol absorption are examined. First, we have
examined the frequency distribution of absorption Ångström exponent ($\alpha_{abs}$, derived from the linear
fit on log-log scale between corresponding absorption coefficients to aethalometer wavelengths) in
Figure 14; separately for the mixed layer (ML, below 1.5 km) and above ($\geq 2$ km). The frequency
distribution of $\alpha_{abs}$ reveals a clear shift towards lower values as we move from JDR to BBR, both
within the ML and above, even though the values of $\alpha_{abs}$ lying mostly between 1 and 1.5. Based on
laboratory studies and field investigations, it has been shown that the higher values of $\alpha_{abs}$ ($\sim 2$) are
representative of biomass burning emissions, while the values $\sim 1$ are indicative of fossil fuel
combustions (Kirchstetter et al., 2004). The values of $\alpha_{abs}>1$ is indicative of the presence of biomass-
burning, whose relative abundance increase with the steepness of the absorption spectra, as has been
reported elsewhere from the laboratory experiments (Hopkins et al., 2007).
Examining Figure 14 in the above light, it emerges that significant contribution of BC from fossil
fuel combustions mixed with biomass burning origin prevails at higher altitudes over BBR, while
the association between the two decreases abruptly from ML to higher height at VNS. The consistent
higher values of BC in the column associated with the values of $\alpha_{abs}$ lying between 1 and 1.5 can
also be due to the aging of BC at higher heights, during which BC mixes with other species and its
angstrom exponent increases, as the spectral dependence of absorption steepens when BC (even
though its source could be fossil fuel) is coated with a concentric shell of weakly absorbing material
(Gogoi et al., 2017). Further investigations are needed in this direction.
3.4 Inter-seasonal variability: a case study at JDR
The spatial variation of the altitude profiles of $N_{AC}$, Dg, $F_{MC}$ and $F_{BC}$ across the IGP hints to several
possible implications of their direct and indirect effects. Altitudinal increases in the values of Dg
and $F_{MC}$ along with depolarization ratios are indicative of the presence of dust (> 4 μm) in the lower
free troposphere, which is known to produce long-wave (warming) radiative effect (Miller et al.,
2006; Tegen and Lacis, 1996). Conversely, significant abundance of accumulation mode aerosols,
in general, might contribute significantly to scattering. For example, a clear seasonal change in the
vertical profiles of $N_{AC}$ is noticeable at JDR, changing of the much steeper variation (vertically) in
winter (as reported by Gogoi et al., 2019) to a near-steady one during just prior to the onset of
monsoon (Figure 15). Based on air-borne measurements during SWAAMI - RAWEX, Vaishya et
al., (2018) have reported that the values of SSA at west IGP varied between 0.935 (at 530) in spring
to 0.84 (at 530 nm) during prior to onset of monsoon, indicating a seasonal change in the aerosol
type and consequently their optical properties.
To examine the role of the dynamical processes at different seasons, we have shown the profiles of
vertical velocity (in pressure coordinates from 1000 hPa to 100 hPa) in Figure 16. These are obtained
from ERA-interim reanalysis data sets. Here, the positive and negative signs of vertical velocity (ω)
are indicative of updraft (as indicated by -ve values of ω) and downdraft (as indicated by +ve values
of ω). A clear seasonal transformation is seen, with increasingly stronger updrafts dominating over
the IGP from December to June, with the intensity increasing from west to east. In the western IGP
regions, the sign of vertical velocity is seemed to change from December to June, progressively
enhancing the magnitude of deep convection towards the onset of monsoon imparting stronger
vertical dispersion and more homogeneous distribution of aerosols in the column.

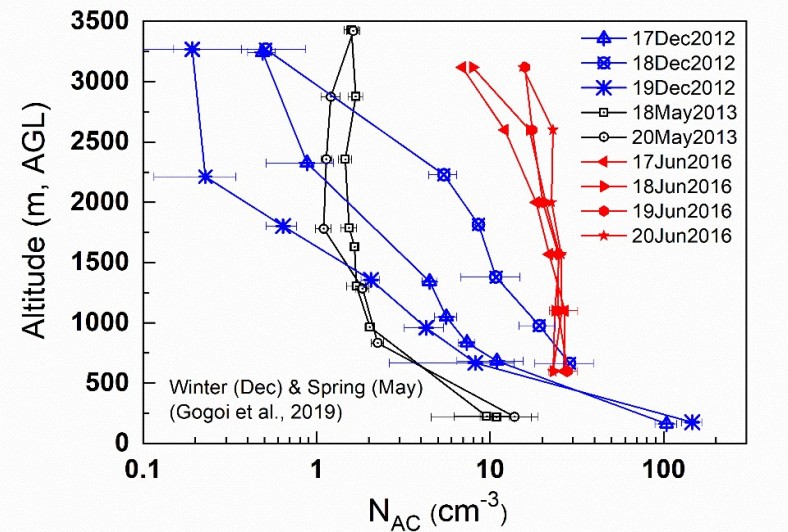


**Figure-15:** Vertical profiles of seasonal mean values of aerosol accumulation and coarse mode
number concentrations ($N_{AC}$) at Jodhpur during winter-2012 (17-19 Dec), spring-2013 (18 and 20
May) and just prior to the onset of monsoon-2016 (17-20 June).

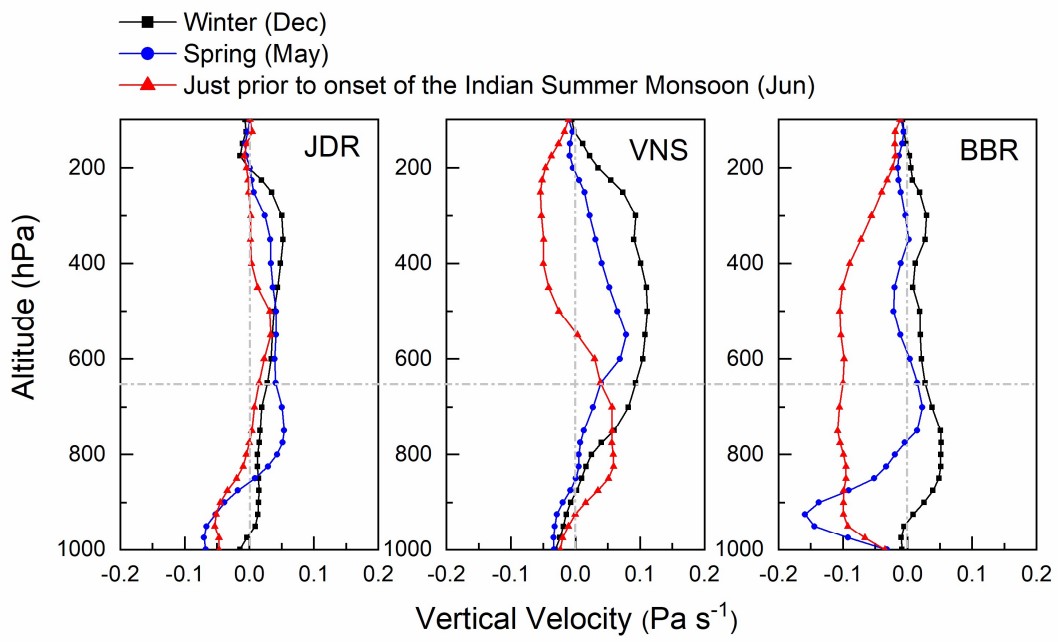


**Figure-16:** Vertical profiles of vertical velocity (Pa s$^{-1}$) over the study locations representing Winter
(December, 2012), Spring (May, 2013) and just prior to onset of the Indian Summer Monsoon (June,
2016) at different pressure levels from 1000 to 100 hPa. The positive and negative values are
indicative of the descending and ascending motions respectively. The horizontal dashed line
indicated the ceiling altitude (~ 3.5 km above ground level) of aircraft measurements while the
vertical dashed lines mark the boundary of vertical velocity (= 0) changing from positive to negative
and vice versa.
Regionally, the seasonal transformation of vertical velocity is more prominent over the eastern IGP
-'BBR', where the magnitude of vertical velocity is consistently higher from surface to upper
tropospheric regions prior to the onset of monsoon. During this period, the head-Bay of Bengal is
known to be one of the regions where deep convection exists (Bhat et al., 2001). Since size
distribution is a dominant factor in determining the direct radiative forcing (Tegen and Lacis, 1996;
Liao and Seinfeld, 1998; Seinfeld et al., 2016), a clear seasonal change in the altitudinal variations
of aerosol type and size distributions associated with distinct transport and convective processes will
have strong radiative impact. Especially the columnar distribution of coarse mode dust and highly
absorbing BC need explicit representations in climate models for accurate understanding of the net
TOA direct radiative forcing. Apart from the direct radiative implications, abundance of coarse mode
dust particles (having sizes larger than critical diameter) and aged BC (coated with hygroscopic
materials) in the lower free troposphere can act as cloud condensation nuclei (CCN) in a
supersaturated environment. Recent studies suggest that mineral aerosols are the dominant ice nuclei
for cirrus clouds (Storelvmo and Herger, 2014).
**4. Summary and Conclusions**
Extensive air-borne measurements of aerosol number-size distribution profiles are carried out, for
the first time across the IGP prior to the onset of Indian summer monsoon as part of SWAAMI -
RAWEX. Collocated measurements of BC profiles are also carried out. The main findings are:
• Aerosol size distribution depicted significant altitudinal variation in the coarse mode regime,
at western IGP (represented by JDR), having highest coarse mode mass fraction (72%) near
the surface; while BC mass fractions ($F_{BC}$) as well as aerosol accumulation and coarse mode
number concentrations ($N_{AC}$) remained nearly steady from surface to the ceiling altitude (~
3.5 km) of the aircraft measurements. However, the pattern was significantly different at
eastern IGP (represented by BBR) transforming to gradually decreasing values of coarse
mode mass concentration ($M_C$) and $N_{AC}$, but with a corresponding increase in the values of
$F_{BC}$ with altitude. At sub-regional scales, BBR depicted higher spatial heterogeneity in the
above aerosol characteristics; while highest homogeneity was observed at JDR.

- Number concentrations showed dominance of accumulation mode near the surface, with the Central IGP station Varanasi (VNS) depicting the highest values $N_{AC}$ ($F_{BC} \sim 15\%$), while the coarse mode remained nearly steady throughout the vertical column.

- Atmospheric heating rate due to BC is highest near the surface at VNS ($\sim 0.81$ K day$^{-1}$), while showing an increasing pattern with altitude at BBR ($\sim 0.35$ K day$^{-1}$) at the ceiling altitude.

- Our measurements, supplemented with information from different space-borne sensors (CATS aboard ISS; OMI) and model results clearly indicated role of mineral dust; both locally generated and advected from the west Asian region, in contributing to the aerosol loading across the IGP, especially at free-tropospheric altitudes. The vertical extents of these layers reached as high as 5 km during the period of observation.

**Data availability**

Details of aircraft data used in this manuscript and the point of contact are available at http://spl.gov.in; "Research Themes"; "Aerosols and Radiative Forcing".

**Authors contributions**

SSB, SKS and KKM conceptualized the experiment and finalized the methodology. SSB, MMG, VJ and AV conducted the measurement on board aircraft. MMG carried out the scientific analysis of the aircraft data and drafted the manuscript with contributions from AV and VJ. KKM, SKS and SSB carried out the review and editing of the manuscript.

**Competing interests**

The authors declare that they have no conflict of interest.

**Acknowledgement**

This study was a part of joint Indo-UK field campaign, South-West Asian Aerosol Monsoon Interactions - Regional Aerosol Warming Experiment (SWAAMI - RAWEX). The aircraft and the flying support were provided by National Remote Sensing Centre (NRSC), Hyderabad. SKS would like to acknowledge J.C. Bose Fellowship awarded to him by SERB-DST. AV was supported by the Department of Science and Technology, Government of India through its INSPIRE Faculty programme. We acknowledge the CATS science team for providing valuable data sets (freely) for scientific applications. The RAWEX project is supported by ISRO (Indian Space Research

Organisation) and the SWAAMI project is supported by MoES (Ministry of Earth Science), Government of India.

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

847                                     *****