# Peer review of "Air-borne in-situ measurements of aerosol size distributions and BC across the IGP during SWAAMI -RAWEX"

_Atmospheric Chemistry and Physics, 2020_

## Referee Comment (RC1) · Anonymous Referee #1 · 6 Apr 2020

Review on 'Air-borne in-situ measurements of aerosol size 1 distributions and 2 BC across the IGP during SWAAMI' by Mukunda Madhab Gogoi1 et al., (ACP-2020-144).

This paper presents altitude profiles of aerosol size distribution and Black carbon obtained through in situ on-board research aircraft as a part of South-West Asian Aerosol Monsoon Interaction (SWAAMI) experiment conducted jointly under Indo-UK project over three distinct locations (Jodhpur, Varanasi, and Bhubaneswar) just prior to the onset of Indian Summer Monsoon. Simultaneous measurements from Cloud Aerosol Transportation System (CATS) on-board International Space Station and OMI measurements are also used as supporting information.

[Figure]

Major results include an increase in coarse mode concentration and coarse mode mass-fraction with increase in altitude across the entire IGP, especially above the well-mixed region. Further authors found increase with altitude in both the mode radii and geometric mean radii of the size distributions. Near the surface the features were specific to the different sub-regions ie., highest coarse mode mass fraction in the western IGP and highest accumulation fraction in the Central IGP with the eastern IGP coming in-between. The elevated coarse mode fraction is attributed to mineral dust load arising from local production as well as due to advection from the west which is further verified using CATS measurements. Existence of a well-mixed BC variation up to the ceiling altitude (3.5 km) is reiterated in this manuscript and match well with those obtained using previous aircraft and balloon platforms.

Results presented in the manuscript are in general unique and apt for the prestigious journal like ACP. Manuscript is written preciously and concise except at few places. The results presents also add new understanding on the size distribution of aerosol concentration in both altitudinal and longitudinally which are very important in understanding their role on precipitation processes besides radiative forcing estimates. Though major part of manuscript is written well, at some places revision is required. Manuscript may be acceptable after satisfactory revising the following.

Major comments/suggestions:

It is not clear from where and how rainfall and relative humidity measurements presented in Figure 2e are obtained. Further they are not discussed at all in the rest of the manuscript. Same for the profiles of temperature presented in figure 2f.

I suggest providing profiles of temperature and relative humidity (if obtained from aircraft) as a separate figure in Supplementary material and add related discussion the manuscript. This information may be useful while dealing with hygroscopic nature of aerosol.

It is not clear what is the source of the data (TPP) presented in Figure 12a. Further (b)

and (c) are inter-changed. Note that SO2 is presented in (c) and NO2 in (b).

Measurements of Black Carbon with Aethalometer: Though authors made correction to the data obtained from Aethalometer, it is not clear how they have taken care of it in the un-pressurized air craft?

It will be good to show heating rates due to BC profiles at these three different regions.

Some discussion is needed on how the results presented in the manuscript are linked with the main objective of the SWAAMI experiment.

Minor issues:

Results presented in Page 4 at lines 90-94 and 111-121 are mostly repeating. Both can be clubbed and rewrite to the point.

Figure 11 caption does not match with the information presented in the figure. I am unable to see (b) Daily profiles of MBC during each of the flight sorties on different days.

Figure 14. Figure caption need to be changed as per the information presented in the Figure (It should be 18 and 20 May but not 19 and 20 May).

I do not see any logic for presented vertical velocities for 2012, 2013 and 2016.

—END—

---

## Referee Comment (RC2) · Anonymous Referee #2 · 12 Apr 2020

The manuscript describes measurements of vertical profiles of size resolved number concentrations using an aerodynamic particle sizer and BC derived from a 7 channel aethalometer from three different going from west to east in the Indo Gangetic Plain (IGP). Measurements were made during an experiment named SWAAMI and the results from this experiment were discussed earlier in a couple of publications (Vaishya et al.,2018; and Govardhan et al.,2019) and probably others. There is a lack of vertical profile data of aerosols over the Indian sub-continent and in particular during the pre-monsoon season when the radiative balance over India and surrounding regions plays large role in driving the monsoon circulation. In that sense this paper is a welcome addition. However, the manuscript feels like the authors have tried to slice and dice

[Figure]

the data in different ways but in the end doesn't seem to add anything new. It may be useful as a document of the data/analysis and I accept the paper with that view, though it often reads like a report than a research paper. The description of the dataset and the outcomes of the analysis is reasonable and there is not a lot that can be said in terms of any technical shortcomings of the arguments presented. I have a few specific comments: Line 462: The authors mention 'soot' emissions as of importance from thermal power plants. I generally assume this is primary fly ash and other suspended particulate matter (heavy metal containing particles). They seem to suggest there is soot and SPM and I am not sure what the distinction is? Line 466: seems to suggest soot is BC. Are there any measurements in the power plant plumes to suggest that BC is a major emission from burning coal in power plants? I haven't come across this in discussions of power plant emissions elsewhere.

Figure 13: The figure shows the large fraction of the measurements with angstrom absorption exponents over values of 1 with median values of 1.3 and significant fraction near 1.5 and over. The authors say this is all fossil fuel emissions. Shouldn't these values of the angstrom absorption coefficient put these in the biomass burning and probably BrC range? Generally what fraction of the absorbing material measured using the technique used here fall in the BrC range as compared to BC?

Figure 11: Either labels on the figure (namely figure (a) and figure(b)) or the title of the figure is either wrong or not clear

Figure 9: The focus of the figure is on values less than 0.3, the scale has just one color below that. It will be better if the color scale is recalibrated and plotted with the scale going from 0 to 0.5.

Line 290: The temperature in the western most location is said to be 40 C. This should make this location have the deepest ABL and is not consistent with the description of ABL depths in lines 238:243

---

## Short Comment (SC1) · By Gogoi et al. · 22 Apr 2020

General comments

Air-borne in-situ measurement of aerosol size distributions and BC across the IGP during SWAAMI

By Gogoi et al.

The paper presents the results of the aircraft in situ measurements carried out at three selected locations in the Indo Gangetic plains (IGP) in the summer of 2016 before the advent of the South West Asian Monsoon to obtain the vertical distribution of compos-

ite aerosols and Black Carbon (BC). The aim of the experiment was to distinguish the characteristics of aerosols in the vertical column from surface to peak aircraft altitude. The location and timing of the experiment was so conceived that it covers the west east cross section of the IGP from the semi-arid desert in the west, the central IGP characterized by significant anthropogenic activity and the east coast location influenced by the marine environment and industrial activity. Some other aspects investigated during the SWAAMI are already in the public domain. The well-planned experiment reveals that in-spite of the known east west heterogeneity in aerosol characteristics in the IGP as unveiled from ground based observations, the coarse mode concentration and coarse mode mass fraction of aerosols representing mainly soil dust increases with increase in altitude across the IGP especially above the well mixed layer. Hence, the mode radii and geometric mean radii of aerosol particles increase with height. The east west heterogeneity is mainly restricted to within the boundary layer e.g. the highest coarse mode mass fraction (of the total aerosol load) is seen in the western IGP and highest accumulation mode mass fraction in the central IGP. The high concentration of coarse mode fraction is attributed to mineral dust loading. Simultaneous International Space Station overpass measurement reveals that dust aerosols reach altitude as high as 5 km in this season. On the contrary, BC mass concentration show very little altitude variation upto the aircraft top height. The results so obtained are new, and significant from the point of view of aerosol-radiation interaction and aerosol-cloud interaction. It also establishes unequivocally for the first time the heterogeneity between aerosols within the ABL and free troposphere in the IGP.

In view of the above, I strongly believe that the paper qualifies for publication in the prestigious journal ACP. I recommend publication of the article with minor revisions.

Minor comments/suggestions

1. Though the paper is well written I would suggest a thorough editing of the text by the authors for more clarity at some points and inadvertent grammatical mistakes or overwrite. Also, chronology in references inside the text should be maintained throughout

the text. 2. How the hygroscopic growth under extremely humid conditions as in case of BBS are taken care of in the APS measurement, please specify in the relevant text. 3. The main concern regarding a few figures and their captions as given below. a. Figure 1: Caption please rewrite. It is the AOD 500 nm which is shown in the surface plot upon which the stations are marked. In the figure, the triangles may be identified with the abbreviated station names for better visibility. Authors may rethink about Figure 1, as not much discussion on it is found in the text. Figure 2 is sufficient to represent the site description with the base stations. Otherwise, a few lines on AOD distribution may be added in the text based on Figure 1. b. Rewrite the figure caption 4. c. Figure 5: Replace . . . 'eastern part of India by '. . . . . .eastern part of IGP'. d. Figure 8, Caption please delete 'distinct': also replace 200 by 2000 e. Figure 11b replace MBC by FBC, perhaps there is some confusion with Fig. S2. f. Figure 12. Please zoom in the areas between the ellipse for clarity of the aircraft tracks. Also interchange b & c in the caption. g. Figure 15 What is indicated by the vertical dashed line? Please mention in the caption. 4. In continuation to comment 1, following are a few suggestions in the text. a. Line 90: correct ". . ..various aerosol properties" as ". . ..various aerosol parameters". b. Line 93: modify as . . ..(Bhubaneswar (BBR),the industrialized coastal location in the eastern end of the IGP. c. Line 90-95 can be merged with Line no 111-120 and figure 1 can be shifted to this section. d. Line: 96-99: Please rewrite the sentence e. Line 103-110: can be placed at the end of this section or can be shifted to section 2.2 f. Line 138-146: Should be rewritten and placed at Line 131. g. Section 2.4 should be merged with section 2.2 or put before section 2.3. h. Line 287-292: See if these sentences are more appropriate to place in previous paragraph (after Line 271) i. Line 349: '. . . . . . . . . . ..organic carbon being strong absorbers of UV radiation', please check. j. Line 503: Rewrite the sub-section heading

---

## Author Comment (AC1) · 24 May 2020

**Response to Reviewer-1**

This paper presents altitude profiles of aerosol size distribution and Black carbon obtained through in situ on-board research aircraft as a part of South-West Asian Aerosol Monsoon Interaction (SWAAMI) experiment conducted jointly under Indo-UK project over three distinct locations (Jodhpur, Varanasi, and Bhubaneswar) just prior to the onset of Indian Summer Monsoon. Simultaneous measurements from Cloud Aerosol Transportation System (CATS) on-board International Space Station and OMI measurements are also used as supporting information.

Major results include an increase in coarse mode concentration and coarse mode mass-fraction with increase in altitude across the entire IGP, especially above the well-mixed region. Further authors found increase with altitude in both the mode radii and geometric mean radii of the size distributions. Near the surface the features were specific to the different sub-regions ie., highest coarse mode mass fraction in the western IGP and highest accumulation fraction in the Central IGP with the eastern IGP coming in-between. The elevated coarse mode fraction is attributed to mineral dust load arising from local production as well as due to advection from the west which is further verified using CATS measurements. Existence of a well-mixed BC variation up to the ceiling altitude (3.5 km) is reiterated in this manuscript and match well with those obtained using previous aircraft and balloon platforms.

Results presented in the manuscript are in general unique and apt for the prestigious journal like ACP. Manuscript is written preciously and concise except at few places. The results present also add new understanding on the size distribution of aerosol concentration in both altitudinal and longitudinally which are very important in understanding their role on precipitation processes besides radiative forcing estimates. Though major part of manuscript is written well, at some place revision is required. Manuscript may be acceptable after satisfactory revising the following.

**We appreciate the summary evaluation of the reviewer and agree to the observations. Following the valuable comments and fruitful suggestions for improving the quality of the manuscript, we have revised it incorporating the review comments of all the reviewers. Our point wise response to each of the comment is given below in bold letters, below the respective comments.**

**Major comments/suggestions:**

It is not clear from where and how rainfall and relative humidity measurements presented in Figure 2e are obtained. Further they are not discussed at all in the rest of the manuscript. Same for the profiles of temperature presented in figure 2f.

I suggest providing profiles of temperature and relative humidity (if obtained from aircraft) as a separate figure in Supplementary material and add related discussion the manuscript. This information may be useful while dealing with hygroscopic nature of aerosol.

**Response: The values of surface meteorological parameters were obtained from the meteorological observations at the respective airports during the period of flight operations. In addition, ambient temperatures at different altitude levels of the atmosphere were obtained from the aircraft sensor. However, we did not have dedicated meteorological sensors and data loggers aboard for continuous recording of ambient RH and T. Hence, we are unable to show the profiles. In view of this, we have modified Fig-2 in the revised manuscript and kept the relevant meteorological information (as numerical values) in the appropriate places.**

It is not clear what is the source of the data (TPP) presented in Figure 12a. Further (b) and (c) are inter-changed. Note that SO2 is presented in (c) and NO2 in (b).

**Response: Thanks for the suggestion. We have included the source of data (https://www.ntpc.co.in/en/power-generation/coal-based-power-stations) used to geo-locate the coal based TPP distributed over the Indian region. The inadvertent error in the figure caption (Figure-13-R1) is also corrected in the revised manuscript.**

Measurements of Black Carbon with Aethalometer: Though authors made correction to the data obtained from Aethalometer, it is not clear how they have taken care of it in the un-pressurized air craft?

**Response: We have elaborated the discussion (as given below) on the estimation of true BC concentrations from the unpressurised operation of aethalometer in the aircraft in the revised manuscript.**

**Line nos. 189-204: "In the present study, BC mass concentrations were obtained at 1-minute interval by operating the aethalometer at 50% of the maximum attenuation and a standard mass flow rate of 2 LPM corresponding to standard temperature ($T_O$, 293 K) and pressure ($P_O$, 1013 hPa). As the unpressurised aircraft climbed higher, the instrument experienced ambient pressure (P) and temperature (T). In order to maintain the set mass flow, the pumping speed of the instrument was automatically increased (through internal program) to aspire more volume of air. However, the volume of air aspirated at ambient pressure and temperature requires to be corrected to standard atmospheric condition for the actual estimate of BC (Moorthy et al., 2004). Thus, the actual volume of air aspirated by the Aethalometer at different atmospheric level is,**

$$V = V_o \frac{P_o T}{P T_o}$$

**Thus, true BC mass concentration ($M_{BC}$) is**

$$M_{BC} = M_{BC}^* \left[\frac{P_o T}{P T_O}\right]^{-1} \qquad\qquad (1)$$

**Here, $M_{BC}^*$ is the instrument measured raw mass concentration of BC at ambient pressure and temperature."**

It will be good to show heating rates due to BC profiles at these three different regions. Some discussion is needed on how the results presented in the manuscript are linked with the main objective of the SWAAMI experiment.

**Response: We comply with suggestion with thanks and have included the heating rate profiles due to BC at all the three distinct regions of the IGP. The methodology of deriving BC forcing and atmospheric heating rate due to BC is included in the revised manuscript as given below:**

**Line nos. 469-504: "To quantify the climatic implications of BC, the heating rate profiles of BC are examined based on the estimation of shortwave direct radiative forcing (DRF) due to BC alone. The DRF due to BC represents the difference between the DRF for aerosols with and without the BC component. The in-situ values of scattering ($\sigma_{sca}$) and absorption ($\sigma_{abs}$) coefficients measured on-board the aircraft were used to estimate spectral values of AOD (layer-integrated $\sigma_{sca} + \sigma_{abs}$), single scattering albedo (SSA) and asymmetry parameter (g) for each level, assuming a well-mixed layer**

of 200 m above and below the measurement altitude (details are available in Vaishya et al., 2018). The layer mean values of AOD, SSA and Legendre moments of the aerosol phase function (derived from Henyey–Greenstein approximation) are used as input in the Santa Barbara DISORT Atmospheric Radiative Transfer (SBDART, Ricchiazzi et al., 1998) model to estimate diurnally averaged DRF (net flux with and without aerosols) at the top ($DRF_{TOA}$) and bottom ($DRF_{SUR}$) of each of the layers. The atmospheric forcing ($DRF_{ATM}$) for each of the levels is then estimated as

$$DRF_{ATM} = DRF_{TOA} - DRF_{SUR}$$

In order to estimate the forcing due to BC alone, optical parameters for aerosols were deduced again. For this, values of $\sigma_{abs}$ were segregated to the contributions by BC ($\sigma_{BC}$) and OC ($\sigma_{OC}$), where $\sigma_{BC}$ were estimated following inverse wavelength dependence of BC (e.g., Vaishya et al., 2017). Based on this, a new set of AOD and SSA for BC-free atmosphere is calculated and fed into SBDART for estimating $DRF_{ALL-BC}$ without the BC component. Thus, DRF due to BC is

$$DRF_{BC} = DRF_{ALL} - DRF_{ALL-BC}$$

Here, $DRF_{ALL}$ represents forcing due to all the aerosol components, including BC. Change in AOD from total to BC-free atmosphere is not significant (< 2%), whereas SSA changes to a greater extent which actually participates in the $DRF_{BC}$ estimation.

[Figure]

Figure-12: Vertical profiles of atmospheric heating rate due to BC (solid lines) and composite (dashed lines) aerosols for the regions of the IGP: (a) JDR in western IGP, (b) VNS in central IGP and (c) BBR in eastern IGP. Data for the composite heating rate profiles are from Vaishya et al., 2018.

The vertical profiles of atmospheric heating rate (HR, estimated based on the atmospheric pressure difference between top and bottom of each layer and aerosol induced forcing in that layer) due to BC alone shows (Figure-12) maximum influence of BC in trapping the SW-radiation at VNS, followed by BBR and JDR. Interestingly, the altitudinal profiles of heating rate are distinctly different over the regions, BBR showing an increase with altitude, while VNS shows the opposite pattern with

**maximum heating (~ (~ 0.81 K day$^{-1}$) near the surface. Enhanced heating at 500-2000 m altitude is seen at JDR. These results indicate the dominant role of absorbing aerosols near the surface at VNS, while the atmospheric perturbation due to elevated layers of absorbing aerosols is conspicuous at BBR (HR ~ 0.35 K day$^{-1}$ at the ceiling altitude). The column integrated values of atmospheric forcing due to BC alone are 7.9 Wm$^{-2}$, 14.3 Wm$^{-2}$ and 8.4 Wm$^{-2}$ at JDR, VNS and BBR respectively."**

**Regarding the linkage of the results to the objectives of SWAAMI, we add (line nos. 78-88) the following.**

**"The information on aerosol size distribution is important for accurately describing the phase function, which describes the angular variation of the scattered intensity. The knowledge of its vertical variation would thus improve the accuracy of ARF estimation and hence heating rates. Such information is virtually non-existing over this region. Further, the knowledge of the variation of size distribution with altitude would be useful better understanding the aerosol-cloud interactions and CCN characteristics, during the evolving and active phase of the Indian monsoon. This was among the important information aimed to be obtained under SWAAMI - RAWEX (https://gtr.ukri.org/projects?ref=NE%2FL013886%2F1 and http://www.spl.gov.in/SPL/index.php/arfs-research/field-campaigns/asfasf) - a joint Indo-UK field experiment involving airborne measurements using Indian and UK aircrafts during different phases of the Indian monsoon, right from just prior to the onset of monsoon (i.e. in the beginning of June)."**

**Minor issues:**

Results presented in Page 4 at lines 90-94 and 111-121 are mostly repeating. Both can be clubbed and rewrite to the point.

**Response: Complied with.**

Figure 11 caption does not match with the information presented in the figure. I am unable to see (b) Daily profiles of MBC during each of the flight sorties on different days.

**Response: Sorry for the oversight. We have corrected the figure caption in the revised manuscript.**

Figure 14. Figure caption need to be changed as per the information presented in the Figure (It should be 18 and 20 May but not 19 and 20 May).

**Response: Complied with.**

I do not see any logic for presented vertical velocities for 2012, 2013 and 2016.

**Response: We have shown the vertical velocities to support the role of changing dynamical processes during three distinct seasons (Winter, represented by December), Spring (represented by May) and just to prior to onset of Indian Summer Monsoon (represented by June), which very well support the vertical profiles of BC at the respective seasons. We have made this clear in the legends of the figure in the revised manuscript.**

[Figure]

**Figure-16: Vertical profiles of vertical velocity (Pa s⁻¹) over the study locations representing Winter (December, 2012), Spring (May, 2013) and just prior to the onset of the Indian Summer Monsoon (June, 2016) at different pressure levels from 1000 to 100 hPa. The positive and negative values are indicative of the descending and ascending motions respectively. The horizontal dashed line indicated the ceiling altitude (~ 3.5 km above ground level) of aircraft measurements while the vertical dashed lines mark the boundary of vertical velocity (= 0) changing from positive to negative and vice versa.**

**References:**

**Moorthy, K. K., Babu, S. S., Sunilkumar, S. V., Gupta, P. K., and Gera, B. S.: Altitude profiles of aerosol BC, derived from aircraft measurements over an inland urban location in India, Geophys. Res. Lett., 31, 1–4, https://doi.org/10.1029/2004GL021336, 2004.**

**Ricchiazzi, P., Yang, S., Gautier, C., and Sowle, D.: SBDART: A research and teaching software tool for plane-parallel radiative transfer in the Earth's atmosphere, B. Am. Meteor. Soc., 79,2101–2114, 1998.**

**Vaishya, A., Singh, P., Rastogi, S., and Babu, S. S.: Aerosol black carbon quantification in the central Indo-Gangetic Plain: Seasonal heterogeneity and source apportionment, Atmos. Res., 185,13–21, https://doi.org/10.1016/j.atmosres.2016.10.001, 2017.**

**Vaishya, A., Babu, S. N. S., Jayachandran, V., Gogoi, M. M., Lakshmi, N. B., Moorthy, K. K., and Satheesh, S. K.: Large contrast in the vertical distribution of aerosol optical properties and radiative effects across the Indo-Gangetic Plain during theSWAAMI–RAWEX campaign, Atmos. Chem. Phys., 18, 17669–17685, https://doi.org/10.5194/acp-18-17669-2018, 2018.**

—END—

---

## Author Comment (AC2) · 24 May 2020

**Response to Reviewer-2**

The manuscript describes measurements of vertical profiles of size resolved number concentrations using an aerodynamic particle sizer and BC derived from a 7 channel aethalometer from three different going from west to east in the Indo Gangetic Plain (IGP). Measurements were made during an experiment named SWAAMI and the results from this experiment were discussed earlier in a couple of publications (Vaishya et al.,2018; and Govardhan et al.,2019) and probably others. There is a lack of vertical profile data of aerosols over the Indian sub-continent and in particular during the pre-monsoon season when the radiative balance over India and surrounding regions plays large role in driving the monsoon circulation. In that sense this paper is a welcome addition. However, the manuscript feels like the authors have tried to slice and dice the data in different ways but in the end doesn't seem to add anything new. It may be useful as a document of the data/analysis and I accept the paper with that view, though it often reads like a report than a research paper. The description of the dataset and the outcomes of the analysis is reasonable and there is not a lot that can be said in terms of any technical shortcomings of the arguments presented.

**We thank the reviewer for the overall evaluation. We have addressed all the comments of the reviewer. Our response to each comment is shown by bold letters, below each comment.**

Specific comments:

Line 462: The authors mention 'soot' emissions as of importance from thermal power plants. I generally assume this is primary fly ash and other suspended particulate matter (heavy metal containing particles). They seem to suggest there is soot and SPM and I am not sure what the distinction is?

Line 466: seems to suggest soot is BC. Are there any measurements in the power plant plumes to suggest that BC is a major emission from burning coal in power plants? I haven't come across this in discussions of power plant emissions elsewhere.

**Response: Sorry for the lack of clarity. We agree with the reviewer that fly ash and SPM are major constituents in TPP emissions. Soot or BC is a major component in SPM. We have elaborated this in the revised manuscript, in addition to highlighting reported literature on BC measurements and characterization near coal burning power plants. The following has been added to the manuscript:**

**Line nos. 505-522: "In this context, we have examined the possible role of the large network of thermal power plants (TPP) over the northern part of India, which is reported to have significant contribution to regional emissions (Singh et al., 2018). These include the emissions of $SO_2$, $NO_x$, $CO_2$, CO, VOC, suspended particulate matter (PM2.5 and PM10, including BC and OC), and other trace metals like mercury (Guttikanda and Jawahar, 2014; Sahu et al., 2017) dispersing over large areas through stacks. Fly ash from coal-fired power plants causes severe environmental degradation in the nearby regions (5-10 km) of TPP (Tiwari et al., 2019). Over the IGP, since more than 70% of the thermal power plants are coal based, emissions of $CO_2$ and $SO_2$ hold more than 47% of the total emission share, while the relative share of PM2.5 and NOx are ~15% and 30% (GAINS, 2012). Based on in-situ measurement of BC, in fixed and transit areas, in close proximity of seven coal-fired TPP in Singrauli (located ~ 700 km north-west of BBR), Singh et al., (2018) have reported that BC concentration reached as high as 200 $\mu g.m^{-3}$ in the transit measurements. The Energy and Resources Institute, India have also reported that emission levels of the carbonaceous (soot or BC) particles are estimated to be around 0.061 gm/kWh per unit of electricity from Indian thermal power plants (Vipradas et al., 2004). Based on emission pathways and ambient PM2.5 pollution over India, Venkataraman et al., (2018) have reported that the**

**types of aerosols emitted from coal burning in thermal power plants and industry in eastern and peninsular India are similar to that of residential biomass combustion. These clearly indicate that TPP are major sources of BC in the atmosphere."**

Figure 13: The figure shows the large fraction of the measurements with angstrom absorption exponents over values of 1 with median values of 1.3 and significant fraction near 1.5 and over. The authors say this is all fossil fuel emissions. Shouldn't these values of the angstrom absorption coefficient put these in the biomass burning and probably BrC range? Generally what fraction of the absorbing material measured using the technique used here fall in the BrC range as compared to BC?

**Response: We are sorry for the lack of clarity on the discussion on Angstrom absorption exponent. We have taken care of the suggestion and revised the discussion on aerosol spectral absorption as given below:**

**Line nos. 550-563: "Based on laboratory studies and field investigations, it has already been shown that the higher values of $\alpha_{abs}$ (~ 2) are representative of BC from biomass burning emissions, while the values ~ 1 are indicative of BC from fossil fuel combustions (Kirchstetter et al., 2004). The values of $\alpha_{abs}$> 1 are indicative of the presence of aerosols from biomass-burning, whose relative abundance increase with the steepness of the spectral absorption spectra, as has been reported elsewhere from the laboratory experiments [Hopkins et al., 2007].**

**Examining Figure 14 in the above light, it emerges that significant contribution of BC from fossil fuel combustions mixed with that from biomass burning origin prevails at higher altitudes over BBR, while the association between the two decreases abruptly from ML to higher heights at VNS. Consistently higher values of BC in the column associated with the values of $\alpha_{abs}$ lying between 1 and 1.5 can also be due to the ageing of BC at higher heights, during which BC mixes with other species and its Angstrom exponent increases, as the spectral dependence of absorption steepens when BC (even though its source could be fossil fuel) is coated with a concentric shell of weakly absorbing material (Gogoi et al., 2017). Further investigations are needed in this direction."**

Figure 11: Either labels on the figure (namely figure (a) and figure(b)) or the title of the figure is either wrong or not clear

**Response: Sorry for the oversight. We have corrected the Figure caption in the revised manuscript.**

Figure 9: The focus of the figure is on values less than 0.3, the scale has just one color below that. It will be better if the color scale is recalibrated and plotted with the scale going from 0 to 0.5.

**Response: Complied with. We have modified the figure in the revised manuscript as shown below.**

[Figure]

Line 290: The temperature in the western most location is said to be 40 C. This should make this location have the deepest ABL and is not consistent with the description of ABL depths in lines 238:243

**Response: We are sorry for this mix-up. The value provided in the manuscript was indicative of the general surface air temperature encountered in that location. The actual values during the flight period, however, were different and this is now provided in the revised manuscript:**

**Line nos. 210-213: "The meteorological conditions across the IGP during the campaign period was generally hot (surface temperature, T ~ 34.7 ± 2.8 °C at JDR, 39 ± 1.9 °C at VNS and 32.8 ± 3.6 °C at BBR at the time of flight take off), with low to moderate relative humidity (RH) at JDR (RH ~ 40%) and VNS (RH ~ 60%)."**

**Line nos. 254-255: "The mean ABL heights are 1.3 ± 0.5 km, 2.3 ±0.5 km and1.4 ± 0.2 km for JDR, VNS, and BBR respectively (Vaishya et al., 2018) at local noon time."**

**References:**

**GAINS, Greenhouse Gas and Air Pollution Interactions and Synergies - South Asia Program. International Institute of Applied Systems Analysis, Laxenburg, Austria, 2010.**

**Guttikunda, S.K. and Jawahar, P.: Atmospheric emissions and pollution from the coal-fired thermal power plants in India, Atmos. Env. 92, 449-460 http://dx.doi.org/10.1016/j.atmosenv.2014.04.057, 2014.**

Hopkins, R. J., Tivanski, A.V., Marten, B.D., and Gilles, M. K., Chemical bonding and structure of black carbon reference materials and individual carbonaceous atmospheric aerosols, J. Aerosol Sci. 38, 573–591, 2007.

Sahu, S.K., Ohara. T., and Beig, G.: The role of coal technology in redefining India's climate change agents and other pollutants Environ. Res. Lett. 12, 105006, https://doi.org/10.1088/1748-9326/aa814a, 2017.

Tiwari, M.K., Bajpai, S., Dewangan. U.K., Environmental Issues in Thermal Power Plants – Review in Chhattisgarh Context, J. Mater. Environ. Sci. 10(11), 1123-1134, 2019.

Venkataraman., C., Brauer, M., Tibrewal, K., Sadavarte, P., Ma, Q., Cohen, A., Chaliyakunne, S., Frostad, J., Klimont, Z., Martin, R.V., Millet, D.B., Philip, S., Walker, K., and Wang, S.: Source influence on emission pathways and ambient PM2:5 pollution over India (2015–2050), Atmos. Chem. Phys., 18, 8017–8039, https://doi.org/10.5194/acp-18-8017-2018, 2018

Vipradas, M., Babu, Y.D., Garud, S., and Kumar, A., Preparation of road map for mainstreaming wind energy in India, TERI project report No. 2002RT66, The Energy and Resources Institute, 2014.

—END—

---

## Author Comment (AC3) · 24 May 2020

**Response to Reviewer-3**

The paper presents the results of the aircraft in situ measurements carried out at three selected locations in the Indo Gangetic plains (IGP) in the summer of 2016 before the advent of the South West Asian Monsoon to obtain the vertical distribution of composite aerosols and Black Carbon (BC). The aim of the experiment was to distinguish the characteristics of aerosols in the vertical column from surface to peak aircraft altitude. The location and timing of the experiment was so conceived that it covers the west east cross section of the IGP from the semi-arid desert in the west, the central IGP characterized by significant anthropogenic activity and the east coast location influenced by the marine environment and industrial activity. Some other aspects investigated during the SWAAMI are already in the public domain. The well-planned experiment reveals that in-spite of the known east west heterogeneity in aerosol characteristics in the IGP as unveiled from ground-based observations, the coarse mode concentration and coarse mode mass fraction of aerosols representing mainly soil dust increases with increase in altitude across the IGP especially above the well mixed layer. Hence, the mode radii and geometric mean radii of aerosol particles increase with height. The east west heterogeneity is mainly restricted to within the boundary layer e.g. the highest coarse mode mass fraction (of the total aerosol load) is seen in the western IGP and highest accumulation mode mass fraction in the central IGP. The high concentration of coarse mode fraction is attributed to mineral dust loading. Simultaneous International Space Station overpass measurement reveals that dust aerosols reach altitude as high as 5 km in this season. On the contrary, BC mass concentration shows very little altitude variation upto the aircraft top height. The results so obtained are new, and significant from the point of view of aerosol-radiation interaction and aerosol-cloud interaction. It also establishes unequivocally for the first time the heterogeneity between aerosols within the ABL and free troposphere in the IGP. In view of the above, I strongly believe that the paper qualifies for publication in the prestigious journal ACP. I recommend publication of the article with minor revisions.

**We appreciate the summary observations of the reviewer and the subsequent recommendation on the manuscript. We have addressed to all the queries/ suggestions raised by the reviewer in the revised manuscript. Our point wise response to each of the comments is given below.**

Minor comments/suggestions

1. Though the paper is well written I would suggest a thorough editing of the text by the authors for more clarity at some points and inadvertent grammatical mistakes or overwrite. Also, chronology in references inside the text should be maintained throughout the text.

   **Complied with. We have made a thorough revision of the text and grammatical issues in the manuscript. We have also taken care of the references, arranging in chronological order.**

2. How the hygroscopic growth under extremely humid conditions as in case of BBS are taken care of in the APS measurement, please specify in the relevant text.

**"The TSI-APS (3321) is suitable for operating at 10 to 90% RH (non-condensing) and 10-40ºC ambient temperature. For BBR it is likely that aerosols grew under high RH conditions but might have also shrunk due to higher instrument temperature as compared to ambient. However, more controlled laboratory experiments are required to ascertain the response of the APS to hygroscopic growth of particles."**

**The above information is included in the revised manuscript, Line nos. 177-181.**

3. The main concern regarding a few figures and their captions as given below.

   a. Figure 1: Caption please re-write. It is the AOD 500 nm which is shown in the surface plot upon which the stations are marked. In the figure, the triangles may be identified with the abbreviated station names for better visibility. Authors may rethink about Figure 1, as not much discussion on it is found in the text. Figure 2 is sufficient to represent the site description with the base stations. Otherwise, a few lines on AOD distribution may be added in the text based on Figure 1.

**Response: Complied with**

   b. Rewrite the figure caption 4.

**Response: Complied with.**

   c. Figure 5: Replace . . . 'eastern part of India by '. . .. . ..eastern part of IGP'.

**Response: Complied with.**

   d. Figure 8, Caption please delete 'distinct': also replace 200 by 2000

**Response: Complied with.**

   e. Figure 11b replace MBC by FBC, perhaps there is some confusion with Fig. S2.

**Response: Complied with. The figure caption is corrected.**

   f. Figure 12. Please zoom in the areas between the ellipse for clarity of the aircraft tracks. Also interchange b & c in the caption.

**Response: A zoom in view of the flight tracks is shown (Figure-13R1) in the revised manuscript. Figure caption is corrected.**

   g. Figure 15 What is indicated by the vertical dashed line? Please mention in the caption.

**Response: Complied with. Thanks for pointing out the error on the vertical lines, which are properly set in the revised figure (Figure-16R1) and details are included in the caption.**

   4. *In continuation to comment 1, following are a few suggestions in the text.*

   a. Line 90: correct ". . .various aerosol properties" as ". . .various aerosol parameters".

**Response: Complied with.**

   b. Line 93: modify as . . . (Bhubaneswar (BBR), the industrialized coastal location in the eastern end of the IGP.

**Response: This sentence is removed as was repeating with the sentence in line no 111-125.**

   c. Line 90-95 can be merged with Line no 111-120 and figure 1 can be shifted to this section.

**Response: Lines 90-95 is modified considering the repeating information with lines 111-120. Figure-1 is shifted to this section.**

   d. Line: 96-99: Please rewrite the sentence

**Response: Complied with, the sentence is modified.**

   e. Line 103-110: can be placed at the end of this section or can be shifted to section 2.2

**Response: Complied with, the lines are shifted to section 2.2**

f. Line 138-146: Should be rewritten and placed at Line 131.

**Response: Complied with. These lines are rewritten and shifted as suggested.**

**Line nos. 127-134: " Figure-2a shows the actual dates of onset of the monsoon at different parts of India in 2016. As can be seen from the figure, despite a delayed onset at the southern tip of India, monsoon advanced fast in to the central/northern parts of India. Yet, all the flight sorties from the respective base stations were completed ahead of the advent of monsoon to that station. At the eastern IGP, the aircraft sorties were made from 'BBR' before the onset of monsoon over India; at 'VNS', the flights were conducted while monsoon advanced only to the central peninsula. The final sets of sorties were conducted at 'JDR' when the monsoon covered most of the central and eastern part of India, but yet to progress towards northwestern parts."**

g. Section 2.4 should be merged with section 2.2 or put before section 2.3.

**Response: Complied with.**

h. Line 287-292: See if these sentences are more appropriate to place in previous paragraph (after Line 271)

**Response: Complied with.**

i. Line 349: '. . .. . .. . .. . .. organic carbon being strong absorbers of UV radiation', please check.

**Response: Complied with. We have modified the sentence as**

**Line nos. 357-359: " Higher values of AAOD at 388 nm are indicative of the presence of dust or biomass burning aerosols. This is because absorption by dust and organic carbon from biomass burning sources have strong wavelength dependency with higher absorption at near-UV wavelengths."**

j. Line 503: Rewrite the sub-section heading

**Response: Complied with. We have modified the section heading as "Inter-seasonal variability: a case study at JDR".**

—END—